# OVER-TRAINING WITH MIXUP MAY HURT GENERALIZATION

**Zixuan Liu**[1]*, **Ziqiao Wang**[1]*, **Hongyu Guo**[2,1], **Yongyi Mao** [1]
[1]University of Ottawa [2]National Research Council Canada
`{zliu282, zwang286, hongyu.guo, ymao}@uottawa.ca`

## ABSTRACT

Mixup, which creates synthetic training instances by linearly interpolating random sample pairs, is a simple and yet effective regularization technique to boost the performance of deep models trained with SGD. In this work, we report a previously unobserved phenomenon in Mixup training: on a number of standard datasets, the performance of Mixup-trained models starts to decay after training for a large number of epochs, giving rise to a U-shaped generalization curve. This behavior is further aggravated when the size of original dataset is reduced. To help understand such a behavior of Mixup, we show theoretically that Mixup training may introduce undesired data-dependent label noises to the synthesized data. Via analyzing a least-square regression problem with a random feature model, we explain why noisy labels may cause the U-shaped curve to occur: Mixup improves generalization through fitting the clean patterns at the early training stage, but as training progresses, Mixup becomes over-fitting to the noise in the synthetic data. Extensive experiments are performed on a variety of benchmark datasets, validating this explanation.

## 1 INTRODUCTION

Mixup has empirically shown its effectiveness in improving the generalization and robustness of deep classification models (Zhang et al., 2018; Guo et al., 2019a;b; Thulasidasan et al., 2019; Zhang et al., 2022b). Unlike the vanilla empirical risk minimization (ERM), in which networks are trained using the original training set, Mixup trains the networks with synthetic examples. These examples are created by linearly interpolating both the input features and the labels of instance pairs randomly sampled from the original training set.

Owning to Mixup's simplicity and its effectiveness in boosting the accuracy and calibration of deep classification models, there has been a recent surge of interest attempting to better understand Mixup's working mechanism, training characteristics, regularization potential, and possible limitations (see, e.g.,

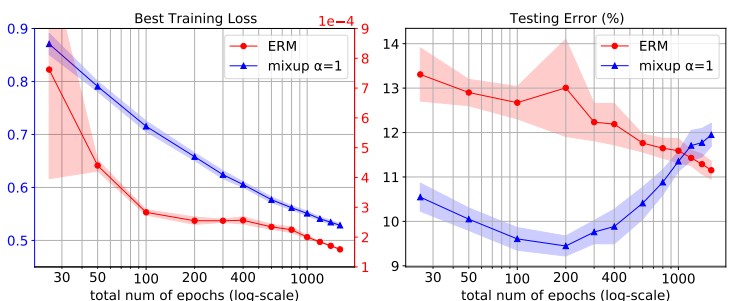

Figure 1: Over-training ResNet18 on CIFAR10.

Thulasidasan et al. (2019), Guo et al. (2019a), Zhang et al. (2021), Zhang et al. (2022b)). In this work, we further investigate the generalization properties of Mixup training.

We first report a previously unobserved phenomenon in Mixup training. Through extensive experiments on various benchmarks, we observe that over-training the networks with Mixup may result in significant degradation of the networks' generalization performance. As a result, along the training

---

*Equal contribution.

epochs, the generalization performance of the network measured by its testing error may exhibit a U-shaped curve. Figure 1 shows such a curve obtained from over-training ResNet18 with Mixup on CIFAR10. As can be seen from Figure 1, after training with Mixup for a long time (200 epochs), both ERM and Mixup keep decreasing their training loss, but the testing error of the Mixup-trained ResNet18 gradually increases, while that of the ERM-trained ResNet18 continues to decrease.

Motivated by this observation, we conduct a theoretical analysis, aiming to better understand the aforementioned behavior of Mixup training. We show theoretically that Mixup training may introduce undesired data-dependent label noises to the synthesized data. Then by analyzing the gradient-descent dynamics of training a random feature model for a least-square regression problem, we explain why noisy labels may cause the U-shaped curve to occur: under label noise, the early phase of training is primarily driven by the clean data pattern, which moves the model parameter closer to the correct solution. But as training progresses, the effect of label noise accumulates through iterations and gradually over-weighs that of the clean pattern and dominates the late training process. In this phase, the model parameter gradually moves away from the correct solution until it is sufficient apart and approaches a location depending on the noise realization.

## 2   RELATED WORK

**Mixup Improves Generalization**   After the initial work of Zhang et al. (2018), a series of the Mixup's variants have been proposed (Guo et al., 2019a; Verma et al., 2019; Yun et al., 2019; Kim et al., 2020; Greenewald et al., 2021; Han et al., 2022; Sohn et al., 2022). For example, *AdaMixup* (Guo et al., 2019a) trains an extra network to dynamically determine the interpolation coefficient parameter $\alpha$. *Manifold Mixup* (Verma et al., 2019) performs linear mixing on the hidden states of the neural networks. Aside from its use in various applications, Mixup's working mechanism and it possible limitations are also being explored constantly. For example, Zhang et al. (2021) demonstrate that Mixup yields a generalization upper bound in terms of the Rademacher complexity of the function class that the network fits. Thulasidasan et al. (2019) show that Mixup helps to improve the calibration of the trained networks. Zhang et al. (2022b) theoretically justify that the calibration effect of Mixup is correlated with the capacity of the network. Additionally, Guo et al. (2019a) point out a "manifold intrusion" phenomenon in Mixup training where the synthetic data "intrudes" the data manifolds of the real data.

**Training on Random Labels, Epoch-Wise Double Descent and Robust Overfitting**   The thought-provoking work of Zhang et al. (2017) highlights that neural networks are able to fit data with random labels. After that, the generalization behavior on corrupted label dataset has been widely investigated (Arpit et al., 2017; Liu et al., 2020; Feng & Tu, 2021; Wang & Mao, 2022; Liu et al., 2022). Specifically, Arpit et al. (2017) observe that neural networks will learn the clean pattern first before fitting to data with random labels. This is further explained by Arora et al. (2019a) where they demonstrate that in the overparameterization regime, the convergence of loss depends on the projections of labels on the eigenvectors of some Gram matrix, where true labels and random labels have different projections. In a parallel line of research, an *epoch-wise* double descent behavior of testing loss of deep neural networks is observed in Nakkiran et al. (2020), shortly after the observation of the *model-wise* double descent (Belkin et al., 2019; Hastie et al., 2022; Mei & Montanari, 2022; Ba et al., 2020). Theoretical works studying the *epoch-wise* double descent are rather limited to date (Heckel & Yilmaz, 2021; Stephenson & Lee, 2021; Pezeshki et al., 2022), among which Advani et al. (2020) inspires the theoretical analysis of the U-sharped curve of Mixup in this paper. Moreover, robust overfitting (Rice et al., 2020) is also another yet related research line, In particular, robust overfitting is referred to a phenomenon in adversarial training that robust accuracy will first increase then decrease after a long training time. Dong et al. (2022) show that robust overfitting is deemed to the early part of epoch-wise double descent due to the *implicit label noise* induced by adversarial training. Since Mixup training has been connected to adversarial training or adversarial robustness in the previous works (Archambault et al., 2019; Zhang et al., 2021), the work of Dong et al. (2022) indeed motivates us to study the label noise induced by Mixup training.

## 3 PRELIMINARIES

Consider a $C$-class classification setting with input space $\mathcal{X} = \mathbb{R}^{d_0}$ and label space $\mathcal{Y} := \{1, 2, \ldots, C\}$. Let $S = \{(\mathbf{x}_i, \mathbf{y}_i)\}_{i=1}^n$ be a training set, where each $\mathbf{y}_i \in \mathcal{Y}$ may also be treated as a one-hot vector in $\mathcal{P}(\mathcal{Y})$, the space of distributions over $\mathcal{Y}$. Let $\Theta$ denote the model parameter space, and for each $\theta \in \Theta$, let $f_\theta : \mathcal{X} \to [0, 1]^C$ denote the predictive function associated with $\theta$, which maps an input feature to a distribution in $\mathcal{P}(\mathcal{Y})$. For any pair $(\mathbf{x}, \mathbf{y}) \in \mathcal{X} \times \mathcal{P}(\mathcal{Y})$, let $\ell(\theta, \mathbf{x}, \mathbf{y})$ denote the loss of the prediction $f_\theta(\mathbf{x})$ with respect to $\mathbf{y}$. The empirical risk of $\theta$ on $S$ is then

$$\hat{R}_S(\theta) := \frac{1}{n} \sum_{i=1}^n \ell(\theta, \mathbf{x}_i, \mathbf{y}_i).$$

When training with Empirical Risk Minimization (ERM), one sets out to find a $\theta^*$ to minimize this risk. It is evident that if $\ell(\cdot)$ is taken as the cross-entropy loss, the empirical risk $\hat{R}_S(\theta)$ is non-negative, where $\hat{R}_S(\theta) = 0$ precisely when $f_\theta(\mathbf{x}_i) = \mathbf{y}_i$ for every $i = 1, 2, \ldots, n$.

In Mixup, instead of using the original training set $S$, the training is performed on a synthetic dataset $\widetilde{S}$ obtained by interpolating training examples in $S$. For a given interpolating parameter $\lambda \in [0, 1]$, let synthetic training set $\widetilde{S}_\lambda$ be defined as

$$\widetilde{S}_\lambda := \{(\lambda\mathbf{x} + (1 - \lambda)\mathbf{x}', \lambda\mathbf{y} + (1 - \lambda)\mathbf{y}') : (\mathbf{x}, \mathbf{y}) \in S, (\mathbf{x}', \mathbf{y}') \in S\} \tag{1}$$

The optimization objective, or the "Mixup loss", is then

$$\mathbb{E}_\lambda \hat{R}_{\widetilde{S}_\lambda}(\theta) := \mathbb{E}_\lambda \frac{1}{|\widetilde{S}_\lambda|} \sum_{(\tilde{\mathbf{x}}, \tilde{\mathbf{y}}) \in \widetilde{S}_\lambda} \ell(\theta, \tilde{\mathbf{x}}, \tilde{\mathbf{y}})$$

where the interpolating parameter $\lambda$ is drawn from a symmetric Beta distribution, $\text{Beta}(\alpha, \alpha)$. The default option is to take $\alpha = 1$. In this case, the following can be proved.

**Lemma 3.1.** *Let $\ell(\cdot)$ be the cross-entropy loss, and $\lambda$ is drawn from $\text{Beta}(1, 1)$ (or the uniform distribution on $[0, 1]$). Then for all $\theta \in \Theta$ and for any given training set $S$ that is balanced,*

$$\mathbb{E}_\lambda \hat{R}_{\widetilde{S}_\lambda}(\theta) \geq \frac{C - 1}{2C},$$

*where the equality holds if and only if $f_\theta(\tilde{\boldsymbol{x}}) = \tilde{\boldsymbol{y}}$ for each synthetic example $(\tilde{\boldsymbol{x}}, \tilde{\boldsymbol{y}}) \in \widetilde{S}_\lambda$.*

For 10-class classification tasks, the bound has value 0.45. Then only when the Mixup loss approaches this value, the found solution is near a true optimum (for models with adequate capacity).

## 4 EMPIRICAL OBSERVATIONS

We conduct experiments using CIFAR10, CIFAR100 and SVHN using ERM and Mixup respectively. For each of the datasets, we have adopted both the original dataset and some balanced subsets obtained by downsampling the original data for certain proportions. SGD with weight decay is used. At each epoch, we record the minimum training loss up to that epoch, as well as the testing accuracy at the epoch achieving the minimum training loss. Results are obtained both for training with data augmentation and for training without. More experimental details are given in Appendix A.

### 4.1 RESULTS ON OVER-TRAINING WITHOUT DATA AUGMENTATION

For CIFAR10 and SVHN, ResNet18 is used, and both the original training sets and subsets with 30% of the original data are adopted. Training is performed for up to 1600 epochs for CIFAR10, and the results are shown in Figure 2. For both the 30% dataset and the full dataset, we see clearly that after some number of epochs (e.g, epoch 200 for the full dataset), the testing accuracy of the Mixup-trained network starts decreasing and this trend continues. This confirms that over-training with Mixup hurts the network's generalization. One would observe a U-shaped curve, as shown in Figure 1 (right), if we were to plot testing error and include results from earlier epochs. Notably, this

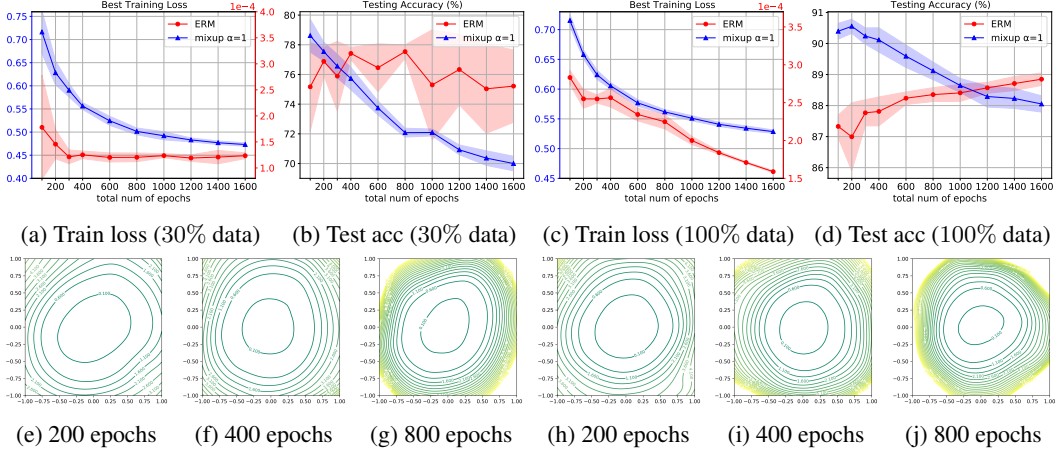

(a) Train loss (30% data)    (b) Test acc (30% data)    (c) Train loss (100% data)    (d) Test acc (100% data)

(e) 200 epochs    (f) 400 epochs    (g) 800 epochs    (h) 200 epochs    (i) 400 epochs    (j) 800 epochs

Figure 2: Training ResNet18 on CIFAR10 training set (100% data and 30% data) without data augmentation. Top row: training loss and testing accuracy for ERM and Mixup. Bottom row: loss landscape of the Mixup-trained ResNet18 (where "loss" refers to the empirical risk on the real data) at various training epochs; left 3 figures are for the 30% CIFAR10 dataset, and the right 3 are for the full CIFAR10 dataset; visualization follows Li et al. (2018)

phenomenon is not observed in ERM. We also found that over-training with Mixup tends to force the network to learn a solution located at the sharper local minima on the loss landscape, a phenomenon correlated with degraded generalization performance (Hochreiter & Schmidhuber, 1997; Keskar et al., 2016). The results of training ResNet18 on SVHN is presented in Appendix B.1.

ResNet34 is used for the more challenging task CIFAR100. This choice allows Mixup to drive its loss to lower values, closer to the lower bound given in Lemma 3.1. In this case, we only use the original training set, since downsampling CIFAR100 appears to result high variances in the testing performance. Training is performed for up to 1600 epochs. The results are plotted in Figure 3. The results again confirm that over-training with Mixup hurts the generalization capability of the learned model. A U-shaped testing loss curve (obtained from a single trial) is also observed in Figure 3c. Additional results of training ResNet34 on CIFAR10 and SVHN are provided in Appendix B.1.

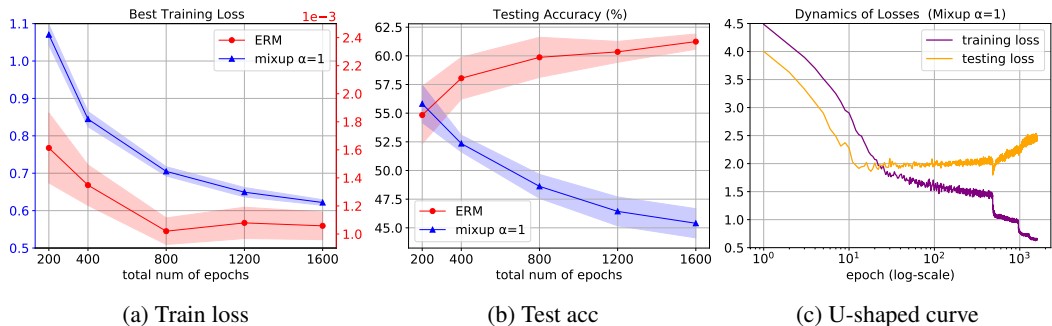

(a) Train loss    (b) Test acc    (c) U-shaped curve

Figure 3: Training loss, testing accuracy and a U-shaped testing loss curve (subfigure (c), yellow) of training ResNet34 on CIFAR100 (100% training data) without data augmentation.

## 4.2 RESULTS ON OVER-TRAINING WITH DATA AUGMENTATION

The data augmentation methods include "random crop" and "horizontal flip" are applied to training on CIFAR10 and CIFAR100. We train ResNet18 on 10% of the CIFAR10 training set for up to 7000 epochs. The results are given in Figures 4a and 4b. In this case, the Mixup-trained model also produces a U-shaped generalization curve. However, while the dataset is downsampled to a lower

proportion, the turning point of the U-shaped curve nevertheless comes much later compared to the previous experiments where data augmentation is not applied on CIFAR10.

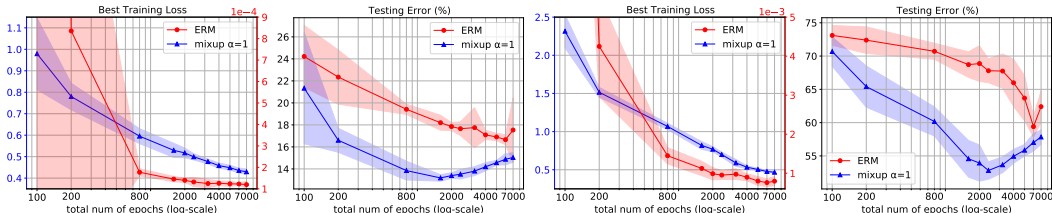

(a) Train loss (CIFAR10)  (b) Test error (CIFAR10)  (c) Train loss (CIFAR100)  (d) Test error (CIFAR100)

Figure 4: (a),(b): Training losses and testing errors of over-training ResNet18 on 10% of the CIFAR10 training set with data augmentation. (c),(d): Training losses and testing errors of over-training ResNet34 on 10% of the CIFAR100 training set with data augmentation.

The results of over-training ResNet34 on 10% of the CIFAR100 training set for up to 7000 epochs are given in Figures 4c and 4d, where similar phenomenons are observed.

## 5 THEORETICAL EXPLANATION

### 5.1 MIXUP INDUCES LABEL NOISE

We will use the capital letters $X$ and $Y$ to denote the random variables representing the input feature and output label, while reserving the notations **x** and **y** to denote their respective realizations. In particular, we consider each true label **y** is as a token, in $\mathcal{Y}$, not a one-hot vector in $\mathcal{P}(\mathcal{Y})$. Let $P(Y|X)$ be the ground-truth conditional distribution of the label $Y$ given input feature $X$. For simplicity, we also express $P(Y|X)$ as a vector-valued function $f : \mathcal{X} \to \mathbb{R}^C$, where $f_j(\mathbf{x}) \triangleq P(Y = j|X = \mathbf{x})$ for each dimension $j \in \mathcal{Y}$.

For simplicity, we consider Mixup with a fixed $\lambda \in [0, 1]$; extension to random $\lambda$ is straight-forward. Let $\widetilde{X}$ and $\widetilde{Y}$ be the random variables corresponding to the synthetic feature and synthetic label respectively. Then $\widetilde{X} \triangleq \lambda X + (1 - \lambda)X'$. Let $P(\widetilde{Y}|\widetilde{X})$ be the conditional distribution of the synthetic label conditioned on the synthetic feature, induced by Mixup, namely, $P(\widetilde{Y} = j|\widetilde{X}) = \lambda f_j(X) + (1-\lambda)f_j(X')$ for each $j$. Then for a synthetic feature $\widetilde{X}$, there are two ways to assign to it a hard label. The first is based on the ground truth, assigning $\widetilde{Y}_{\mathrm{h}}^* \triangleq \arg\max_{j \in \mathcal{Y}} f_j(\widetilde{X})$. The second is based on the Mixup-induced conditional $P(\widetilde{Y}|\widetilde{X})$, assigning $\widetilde{Y}_{\mathrm{h}} \triangleq \arg\max_{j \in \mathcal{Y}} P(\widetilde{Y} = j|\widetilde{X})$. When the two assignments disagree, or $\widetilde{Y}_{\mathrm{h}} \neq \widetilde{Y}_{\mathrm{h}}^*$, we say that the Mixup-assigned label $\widetilde{Y}_{\mathrm{h}}$ is noisy.

**Theorem 5.1.** *For any fixed $X$, $X'$ and $\widetilde{X}$ related by $\widetilde{X} = \lambda X + (1 - \lambda)X'$ for a fixed $\lambda \in [0, 1]$, the probability of assigning a noisy label is lower bounded by*

$$P(\widetilde{Y}_{\mathrm{h}} \neq \widetilde{Y}_{\mathrm{h}}^*|\widetilde{X}) \geq \mathrm{TV}(P(\widetilde{Y}|\widetilde{X}), P(Y|X)) \geq \frac{1}{2} \sup_{j \in \mathcal{Y}} \left| f_j(\widetilde{X}) - [(1 - \lambda)f_j(X) + \lambda f_j(X')] \right|,$$

*where $\mathrm{TV}(\cdot, \cdot)$ is the total variation (see Appendix D).*

**Remark 5.1.** *This lower bound hints that the label noise induced by Mixup training depends on the distribution of original data $P_X$, the convexity of $f(X)$ and the value of $\lambda$. Clearly, Mixup will create noisy labels with non-zero probability (at least for some $\lambda$) unless $f_j$ is linear for each $j$.*

**Remark 5.2.** *We often consider that the real data are labelled with certainty, i.e., $\max_{j \in \mathcal{Y}} f_j(X) = 1$ and $\sum_{j=1}^{C} f_j(X) = 1$. Then the probability of assigning noisy label to a given synthetic data can be discussed in three situations: i) if $\widetilde{Y}_{\mathrm{h}}^* \notin \{Y, Y'\}$, where $Y$ could be the same with $Y'$, then $\widetilde{Y}$ is a noisy label with probability one; ii) if $\widetilde{Y}_{\mathrm{h}}^* \in \{Y, Y'\}$ where $Y \neq Y'$, then the probability of assigning a noisy label is non-zero and depends on $\lambda$; iii) if $\widetilde{Y}_{\mathrm{h}}^* = Y = Y'$, then $\widetilde{Y}_{\mathrm{h}}^* = \widetilde{Y}$.*

As shown in (Arpit et al., 2017; Arora et al., 2019a), when neural networks are trained with a fraction of random labels, they will first learn the clean pattern and then overfit to noisy labels. In Mixup

training, we in fact create much more data, possibly with noisy labels, than traditional ERM training ($n^2$ for a fixed $\lambda$). Thus, one may expect an improved performance (relative to ERM) in the early training phase, due to the clean pattern in the enlarged training set, but a performance impairment in the later phase due to noisy labels. Specifically if $\widetilde{Y}_{\mathrm{h}}^* \notin \{Y, Y'\}$ happens with a high chance, a phenomenon known as "manifold intrusion" (Guo et al., 2019a), then the synthetic dataset contains too many noisy labels, causing Mixup to perform inferior to ERM.

Theorem 5.1 has implied that, in classification problems, Mixup training induces label noise. Next, we will provide a theoretical analysis using a regression setup to explain that such label noise may result in the U-shape learning curve. The choice of a regression setup in this analysis is due to the difficulty in directly analyzing classification problems (under the cross-entropy loss). Such a regression setting may not perfectly explain the U-shaped curve in classification tasks, we however believe that they give adequate insight illuminating such scenarios as well. Such an approach has been taken in most analytic works that study the behaviour of deep learning. For example, Arora et al. (2019b) uses a regression setup to analyze the optimization and generalization property of overparameterized neural networks. Yang et al. (2020) theoretically analyze the bias-variance trade-off in deep network generalization using a regression problem.

## 5.2 REGRESSION SETTING WITH RANDOM FEATURE MODELS

Consider a simple least squares regression problem. Let $\mathcal{Y} = \mathbb{R}$ and let $f : \mathcal{X} \to \mathcal{Y}$ be the ground-truth labelling function. Let $(\widetilde{X}, \widetilde{Y})$ be a synthetic pair obtained by mixing $(X, Y)$ and $(X', Y')$. Let $\widetilde{Y}^* = f(\widetilde{X})$ and $Z \triangleq \widetilde{Y} - \widetilde{Y}^*$. Then $Z$ can be regarded as noise introduced by Mixup, which may be data-dependent. For example, if $f$ is strongly convex with some parameter $\rho > 0$, then $Z \geq \frac{\rho}{2}\lambda(1-\lambda)||X - X'||_2^2$. Given a synthesized training dataset $\widetilde{S} = \{(\widetilde{X}_i, \widetilde{Y}_i)\}_{i=1}^m$, consider a random feature model, $\theta^T \phi(X)$, where $\phi : \mathcal{X} \to \mathbb{R}^d$ and $\theta \in \mathbb{R}^d$. We will consider $\phi$ fixed and only learn the model parameter $\theta$ using gradient descent on the MSE loss

$$\hat{R}_{\widetilde{S}}(\theta) \triangleq \frac{1}{2m} \left|\left| \theta^T \widetilde{\Phi} - \widetilde{\mathbf{Y}}^T \right|\right|_2^2,$$

where $\widetilde{\Phi} = [\phi(\widetilde{X}_1), \phi(\widetilde{X}_2), \dots, \phi(\widetilde{X}_m)] \in \mathbb{R}^{d \times m}$ and $\widetilde{\mathbf{Y}} = [\widetilde{Y}_1, \widetilde{Y}_2, \dots, \widetilde{Y}_m] \in \mathbb{R}^m$.

For a fixed $\lambda$, Mixup can create $m = n^2$ synthesized examples. Thus it is reasonable to assume $m > d$ (e.g., under-parameterized regime) in Mixup training. For example, ResNet-50 has less than 30 million parameters while the square of CIFAR10 training dataset size is larger than 200 million without using other data augmentation techniques. Then the gradient flow, as shown in Liao & Couillet (2018), is

$$\dot{\theta} = -\eta \nabla \hat{R}_{\widetilde{S}}(\theta) = \frac{\eta}{m} \widetilde{\Phi}\widetilde{\Phi}^T \left( \widetilde{\Phi}^\dagger \widetilde{\mathbf{Y}} - \theta \right), \tag{2}$$

where $\eta$ is learning rate and $\widetilde{\Phi}^\dagger = (\widetilde{\Phi}\widetilde{\Phi}^T)^{-1}\widetilde{\Phi}$ is the Moore–Penrose inverse of $\widetilde{\Phi}^T$ (only possible when $m > d$). Thus, we have the following important lemma.

**Lemma 5.1.** *Let $\theta^* = \widetilde{\Phi}^\dagger \widetilde{\mathbf{Y}}^*$ and $\theta^{\mathrm{noise}} = \widetilde{\Phi}^\dagger \mathbf{Z}$ wherein $\mathbf{Z} = [Z_1, Z_2, \dots, Z_m] \in \mathbb{R}^m$, the ODE in Eq. (2) has the following closed form solution*

$$\theta_t - \theta^* = (\theta_0 - \theta^*)e^{-\frac{\eta}{m}\widetilde{\Phi}\widetilde{\Phi}^T t} + (\mathbf{I}_d - e^{-\frac{\eta}{m}\widetilde{\Phi}\widetilde{\Phi}^T t})\theta^{\mathrm{noise}}. \tag{3}$$

**Remark 5.3.** *Notably, $\theta^* = \widetilde{\Phi}^\dagger \widetilde{\mathbf{Y}}^*$ may be seen as the "clean pattern" of the training data. The first term in Eq. (3) is decreasing (in norm) and vanishes at $t \to \infty$. Thus its role is moving $\theta_t$ towards the clean pattern $\theta^*$, allowing the model to generalize to unseen data. But it only dominates the dynamics of $\theta_t$ in the early training phase. The second term, initially 0, increases with $t$ and converges to $\theta^{\mathrm{noise}}$ as $t \to \infty$. Thus its role is moving $\theta_t$ towards the "noisy pattern" $\theta^* + \theta^{\mathrm{noise}}$. It dominates the later training phase and hence hurts generalization. It is noteworthy that $\theta^* + \theta^{\mathrm{noise}}$ is also the closed-form solution for the regression problem (under Mixup labels). This suggests that the optimization problem associated with the Mixup loss has a "wrong" solution, but it is possible to benefit from only solving this problem partially, using gradient descent without over-training.*

Noting that the population risk at time step $t$ is

$$R_t \triangleq \mathbb{E}_{\theta_t, X, Y} \left|\left| \theta_t^T \phi(X) - Y \right|\right|_2^2,$$

and the true optimal risk is

$$R^* = \mathbb{E}_{X,Y} \left|\left| Y - \theta^{*T}\phi(X) \right|\right|_2^2,$$

we have the following result.

**Theorem 5.2** (Dynamics of Population Risk). *Given a synthesized dataset $\widetilde{S}$, assume $\theta_0 \sim \mathcal{N}(0, \xi^2 I_d)$, $||\phi(X)||^2 \le C_1/2$ for some constant $C_1 > 0$ and $|Z| \le \sqrt{C_2}$ for some constant $C_2 > 0$, then we have the following upper bound*

$$R_t - R^* \le C_1 \sum_{k=1}^{d} \left[ \left(\xi_k^2 + \theta_k^{*2}\right) e^{-2\eta\mu_k t} + \frac{C_2}{\mu_k} \left(1 - e^{-\eta\mu_k t}\right)^2 \right] + 2\sqrt{C_1 R^* \zeta},$$

*where $\zeta = \sum_{k=1}^{d} \max\{\xi_k^2 + \theta_k^{*2}, \frac{C_2}{\mu_k}\}$ and $\mu_k$ is the $k^{\text{th}}$ eigenvalue of the matrix $\frac{1}{m}\widetilde{\Phi}\widetilde{\Phi}^T$.*

**Remark 5.4.** *The additive noise $Z$ is usually assumed as a zero mean Gaussian in the literature of generalization dynamics analysis (Advani et al., 2020; Pezeshki et al., 2022; Heckel & Yilmaz, 2021), but this would be hardly justifiable in this context. The boundness assumption of $Z$ in the theorem can however be easily satisfied as long as the output of $f$ is bounded.*

**Remark 5.5.** *If we further let $\xi = 0$ (i.e. using zero initialization) and assume that the eigenvalues of the matrix $\frac{1}{m}\widetilde{\Phi}\widetilde{\Phi}^T$ are all equal to $\mu$, then the summation part in the bound above can be re-written as $C_1 ||\theta^*||^2 e^{-2\eta\mu t} + (C_2/\mu)(1 - e^{-\eta\mu t})^2$, then it is clear that the magnitude of the curve is controlled by the norm of $\theta^*$, the norm of the representation, the noise level and $\mu$.*

Theorem 5.2 indicates that the population risk will first decrease due to the first term (i.e. $\left(\xi_k^2 + \theta_k^{*2}\right) e^{-2\eta\mu_k t}$) then it will grow due to the existence of label noises (i.e. $\frac{C_2}{\mu_k}\left(1 - e^{-\eta\mu_k t}\right)^2$). Overall, the population risk will be endowed with a U-shaped curve. Notice that the quantity $\eta\mu_k$ plays a key role in the upper bound, the larger id $\eta\mu_k$, the earlier comes the turning point of "U". This may have an interesting application, justifying a multi-stage training strategy where the learning rate is reduced at each new stage. Suppose that with the initial learning rate, at epoch $T$, the test error has dropped to the bottom of the U-curve corresponding to this learning rate. If the learning rate is decreased at this point, then the U-curve corresponding to the new learning rate may have a lower minimum error and its bottom shifted to the right. In this case, the new learning rate allows the testing error to move to the new U-curve and further decay.

## 6 EMPIRICAL VERIFICATION

### 6.1 A TEACHER-STUDENT TOY SETUP

To empirically verify our theoretical results discussed in Section 5.2, we construct a simple teacher-student regression setting. The teacher network is a two-layer neural networks with **Tanh** activation and random weights. It only serves to create training data for the student network. Specifically, the training data is created by drawing $\{X_i\}_{i=1}^n$ i.i.d. from a standard Gaussian $\mathcal{N}(0, I_{d_0})$ and passing them to teacher network to obtain labels $\{Y_i\}_{i=1}^n$.

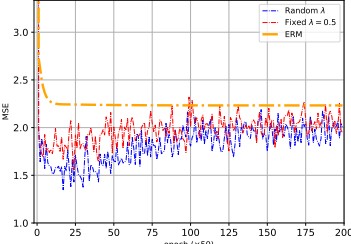

Figure 5: Dynamics of testing loss in the toy example.

The student network is also a two-layer neural network with **Tanh** activation and hidden layer dimension $d = 100$. We fix the parameters in the first layer and only train the second layer using the generated training data. Full-batch gradient descent on the MSE loss is used. For the value of $\lambda$, we consider two cases: a fixed value with $\lambda = 0.5$ and random values drawn from $\text{Beta}(1, 1)$ at each epoch. As a comparison, we also present the result of ERM training in an over-parameterized regime (i.e., $n < d$).

The testing loss dynamics are presented in Figure 5. We first note that Mixup still outperforms ERM in this regression problem, but clearly, only Mixup training has a U-shaped curve while the testing loss of ERM training converges to a constant value. Furthermore, the testing loss of Mixup training is endowed with a U-shaped behavior for both fixed $\lambda = 0.5$ and random $\lambda$ drawn from $\text{Beta}(1, 1)$.

This suggests that our analysis of Mixup in Section 5.2 based on a fixed $\lambda$ is also indicative for more general settings of $\lambda$. Figure 5 also indicates that when $\lambda$ is fixed to 0.5, the increasing stage of the U-shaped curve comes earlier than that of $\lambda$ with $\text{Beta}(1, 1)$. This is consistent with our theoretical results in Section 5.2. That is, owning to the fact that $\lambda$ with the constant value 0.5 for $\lambda$ represents the largest noise level in Mixup, the noise-dominating effect in Mixup training comes earlier.

## 6.2 USING MIXUP ONLY IN THE EARLY STAGE OF TRAINING

In the previous section, we have argued that Mixup training learns "clean patterns" in the early stage of the training process and then overfits the "noisy patterns" in the later stage. Such a conclusion implies that turning of Mixup after a certain number of epochs and returning to standard ERM training may prevent the training from overfitting the noises induced by Mixup. We now present results obtained from such a training scheme on both CIFAR10 and SVHN in Figure 6.

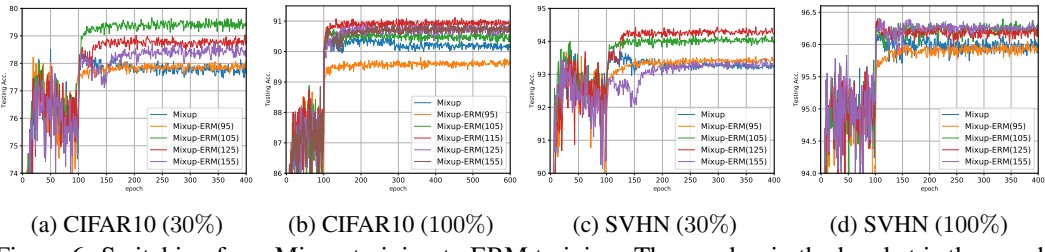

(a) CIFAR10 (30%)     (b) CIFAR10 (100%)     (c) SVHN (30%)     (d) SVHN (100%)

Figure 6: Switching from Mixup training to ERM training. The number in the bracket is the epoch number where we let $\alpha = 0$ (i.e. Mixup training becomes ERM training).

Results in Figure 6 clearly indicate that switching from Mixup to ERM at an appropriate time will successfully avoid the generalization degradation. Figure 6 also suggests that switching Mixup to ERM too early may not boost the model performance. In addition, if the switch is too late, memorization of noisy data may already taken effect, which impact generalization negatively. We note that our results here can be regarded as a complement to (Golatkar et al., 2019), where the authors show that regularization techniques only matter during the early phase of learning.

## 7 FURTHER INVESTIGATION

**Impact of Data Size on U-shaped Curve** In the over-training experiments without data augmentation, although the U-shaped behavior occurs on both $100\%$ and $30\%$ of the original training data for both CIFAR10 and SVHN, we notice that smaller size datasets appear to enable the turning point of the U-shaped curve to arrive earlier. We now corroborate this phenomenon with more experimental results, as shown in Figure 7. In this context, an appropriate data augmentation can be seen as simply expanding the training set with additional clean data. Then the impact of data augmentation on the over-training dynamics of Mixup is arguably via increasing the size of the training set. This explains our observations in Section 4.2 where the turning points in training with data augmentation arrive much later compared to those without data augmentation. Those observations are also consistent with the results in Figure 7.

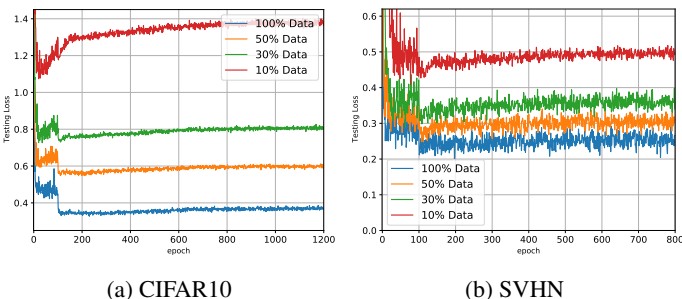

(a) CIFAR10     (b) SVHN

Figure 7: Over-training on different number of samples.

It may be tempting to consider the application of the usual analysis of generalization dynamics from the existing literature (Liao & Couillet, 2018; Advani et al., 2020; Stephenson & Lee, 2021) to the training of Mixup. For example, one can analyze the distribution of the eigenvalues in Theorem 5.2. Specifically, if entries in $\Phi$ are independent identically distributed with zero mean, then in the limit of $d, m \to \infty$ with $d/m = \gamma \in (0, +\infty)$, the eigenvalues $\{\mu_k\}_{k=1}^{d}$ follow the Marchenko-Pastur (MP) distribution (Marčenko & Pastur, 1967),

which is defined as

$$P^{MP}(\mu|\gamma) = \frac{1}{2\pi} \frac{\sqrt{(\gamma_+ - \mu)(\mu - \gamma_-)}}{\mu\gamma} \mathbf{1}_{\mu \in [\gamma_-, \gamma_+]},$$

where $\gamma_{\pm} = (1 \pm \gamma)^2$. Note that the $P^{MP}$ are only non-zero when $\mu = 0$ or $\mu \in [\gamma_-, \gamma_+]$. When $\gamma$ is close to one, the probability of extremely small eigenvalues is immensely increased. From Theorem 5.2, when $\mu_k$ is small, the second term, governed by the noisy pattern, will badly dominate the behavior of population risk and converge to a larger value. Thus, letting $d \ll m$ will alleviate the domination of the noise term in Theorem 5.2. However, it is important to note that such analysis lacks rigor since the columns in $\Phi$ are not independent (two columns might result from linearly combining the same pair of original instances). To apply a similar analysis here, one need to remove or relax the independence conditions on the entries of $\Phi$, for example, by invoking some techniques similar to that developed in Bryson et al. (2021). This is beyond the scope of this paper, and we will to leave it for future study.

**Gradient Norm in Mixup Training Does Not Vanish** Normally, ERM training obtains zero gradient norm at the end of training, which indicates that SGD finds a local minimum. However, We observe that the gradient norm of Mixup training does not converge to zero, as shown in Figure 8.

In fact, gradient norm in the Mixup training even increases until converging to a maximum value, as opposed to zero. When models are trained with ERM on random labels, this increasing trend of gradient norm is also observed in the previous works (Feng & Tu, 2021; Wang & Mao, 2022). Specifically, in Wang & Mao (2022), such increasing behavior is interpreted

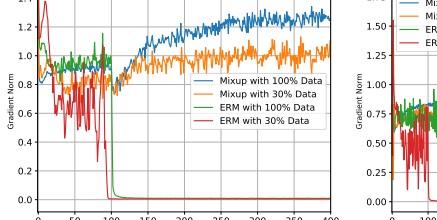

(a) Gradient norm on CIFAR10     (b) Gradient norm on SVHN

Figure 8: Dynamics of gradient norm.

as a sign that the training of SGD enters a "memorization regime", and after the overparameterized neural networks memorize all the noisy labels, the gradient norm (or gradient dispersion in Wang & Mao (2022)) will decrease again until it converges to zero. In Mixup training, since the size of synthetic dataset is usually larger than the number of parameters (i.e., $m > d$), neural networks may not be able to memorize all the noisy labels in this case. Notice that $m$ is much larger than $n^2$ in practice since $\lambda$ is not fixed to a constant.

Notably, although ERM training is able to find a local minimum in the first 130 epochs on CIFAR10, Figure 1 indicates that Mixup training outperforms ERM in the first 400 epochs. Similar observation also holds for SVHN. This result in fact suggests that Mixup can generalize well without converging to any stationary points. Notice that there is a related observation in the recent work of Zhang et al. (2022a), where they show that large-scale neural networks generalize well without having the gradient norm vanish during training. Additionally, by switching Mixup training to ERM training, as what we did in Figure 6, the gradient norm will instantly become zero (see Figure 14 in Appendix B.3). This further justifies that the "clean patterns" are already learned by Mixup trained neural networks at the early stage of training, and the original data may no longer provide any useful gradient signal.

## 8 CONCLUDING REMARKS

We discovered a novel phenomenon in Mixup: over-training with Mixup may give rise to a U-shaped generalization curve. We theoretically show that this is due to the data-dependent label noises introduced to the synthesized data, and suggest that Mixup improves generalization through fitting the clean patterns at the early training stage, but over-fits the noise as training proceeds. The effectiveness of Mixup and the fact it works by only partially optimizing its loss function without reaching convergence, as are validated by our analysis and experiments, seem to suggest that the dynamics of the iterative learning algorithm and an appropriate criteria for terminating the algorithm might be more essential than the loss function or the solutions to the optimization problem. Exploration in the space of iterative algorithms (rather than the space of loss functions) may lead to fruitful discoveries.

ACKNOWLEDGMENTS

This work is supported in part by a National Research Council of Canada (NRC) Collaborative R&D grant (AI4D-CORE-07). Ziqiao Wang is also supported in part by the NSERC CREATE program through the Interdisciplinary Math and Artificial Intelligence (INTER-MATH-AI) project.

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

## A   EXPERIMENTAL SETUPS OF OVER-TRAINING

For any experimental setting ( such as the training dataset and its size, whether ERM or Mixup is used, whether other data augmentation is used, etc.), we define a training "trial" as a training process starting from random initialization to a certain epoch $t$. In each trial, we record the minimum training loss obtained during the entire training process. The testing accuracy of the model's intermediate solution that gives rise to that minimum training loss is also recorded. For different trials we gradually increase $t$ so as to gradually let the model be over-trained. For each $t$, we repeat the trial for 10 times with different random seeds and collect all the recorded results (minimum training losses and the corresponding testing accuracies). We then compute their averages and standard deviations for all $t$'s. These results are eventually used to plot the line graphs for presentation.

For example, Figure 2a illustrates the results of training ResNet18 on 30% CIFAR10 data without data augmentation. The total number of training epochs $t$, as shown on the horizontal axis, is increased from 100 to 1600. For each $t$, each point on its vertical axis represents the average of the recorded training losses from the 10 repeats. The width of the shade beside each point reflects the corresponding standard deviation.

## B   ADDITIONAL EXPERIMENTAL RESULTS

### B.1   ADDITIONAL RESULTS OF OVER-TRAINING WITHOUT DATA AUGMENTATION

Besides CIFAR10, ResNet18 is also used for the SVHN dataset.

Training is performed for up to 1000 epochs for SVHN, since we notice that if we continue training ResNet18 on SVHN after 1000 epochs, the variance of the testing accuracy severely increases. The results are presented in Figure 9. Mixup exhibits a similar phenomenon as it does for CIFAR10. What differs notably is that over-training with ERM on the original SVHN training set appears to also lead to worse test accuracy. However, this does not occur on the 30% SVHN training set[1].

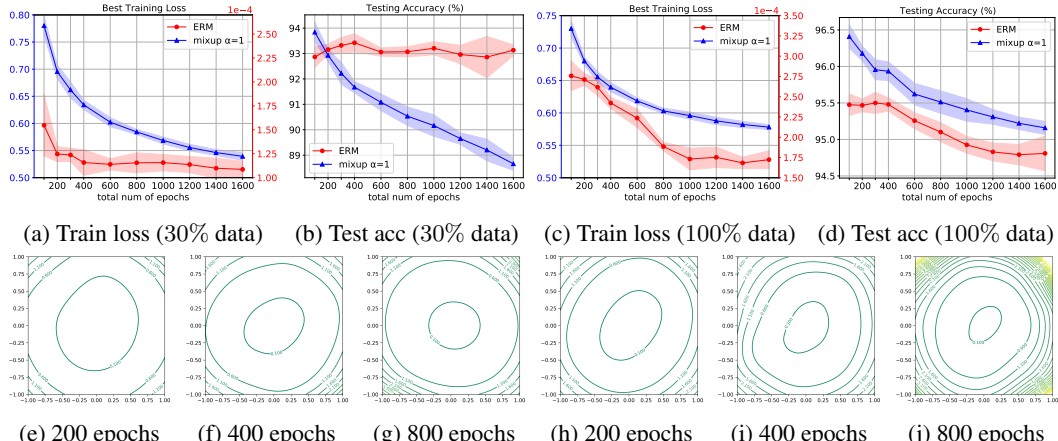

(a) Train loss (30% data)    (b) Test acc (30% data)    (c) Train loss (100% data)    (d) Test acc (100% data)

(e) 200 epochs    (f) 400 epochs    (g) 800 epochs    (h) 200 epochs    (i) 400 epochs    (j) 800 epochs

Figure 9: Results of training ResNet18 on SVHN training set (100% data and 30% data) without data augmentation. Top raw: training loss and testing accuracy for ERM and Mixup. Bottom raw: loss landscape of the Mixup-trained ResNet18 at various training epochs: the left 3 figures are for the 30% SVHN dataset, and the right 3 are for the full SVHN dataset.

As for ResNet34, besides CIFAR100, it is also used for both the CIFAR10 and the SVHN datasets.

---

[1]This might be related to the epoch-wise double descent behavior of ERM training. That is, when over-training ResNet18 on the whole training set with a total of 1000 epochs, the network is still in the first stage of over-fitting the training data, while when over-training the network on 30% of the training set, the network learns faster on the training data due to the smaller sample size, thus it passes the turning point of the double descent curve earlier.

Training is performed on both datasets for in total 200, 400 and 800 epochs respectively. The results for CIFAR10 are shown in Figure 10. For both the 30% dataset and the original dataset, Mixup exhibits a similar phenomenon as it does in training ResNet18 on CIFAR10. The difference is that over-training ResNet34 with ERM let the testing accuracy gradually increase on both the 30% dataset and the original dataset.

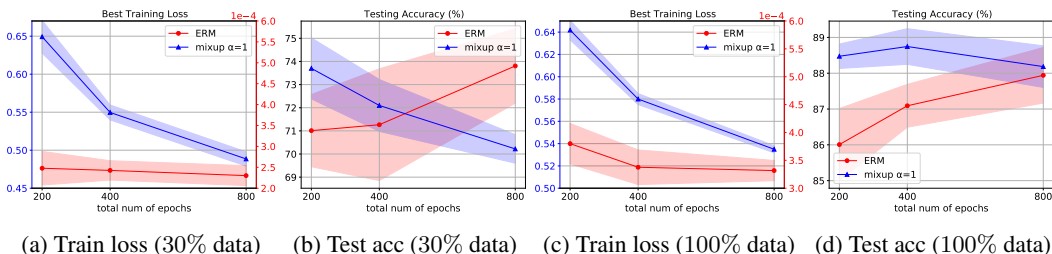

(a) Train loss (30% data)    (b) Test acc (30% data)    (c) Train loss (100% data)    (d) Test acc (100% data)

Figure 10: Results of the recorded training losses and testing accuracies of training ResNet34 on CIFAR10 training set (100% data and 30% data) without data augmentation.

The results for SVHN are shown in Figure 11. These results are also in accordance with those of training ResNet18 on both 30% and 100% of the SVHN dataset.

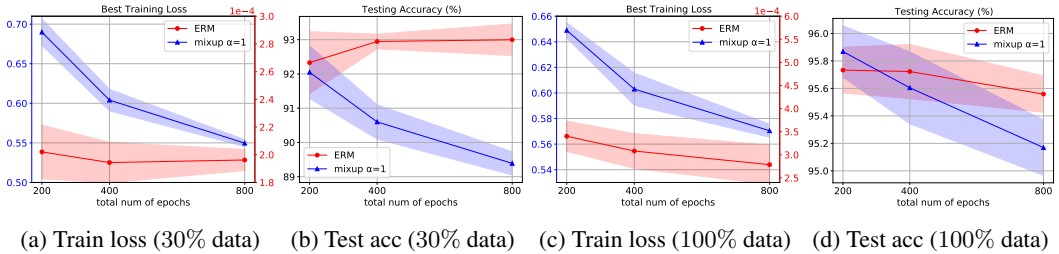

(a) Train loss (30% data)    (b) Test acc (30% data)    (c) Train loss (100% data)    (d) Test acc (100% data)

Figure 11: Results of the recorded training losses and testing accuracies of training ResNet34 on SVHN training set (100% data and 30% data) without data augmentation.

In addition, we have trained VGG16 on the CIFAR10 training set (100% data and 30% data) for up to in total 1600 epochs without data augmentation. The results are provided in Figure 12. In both cases, over-training VGG16 with either ERM or Mixup can gradually reduce the best achieved training loss. However, the testing accuracy of the Mixup-trained network also decreases, while that of the ERM-trained network has no significant change.

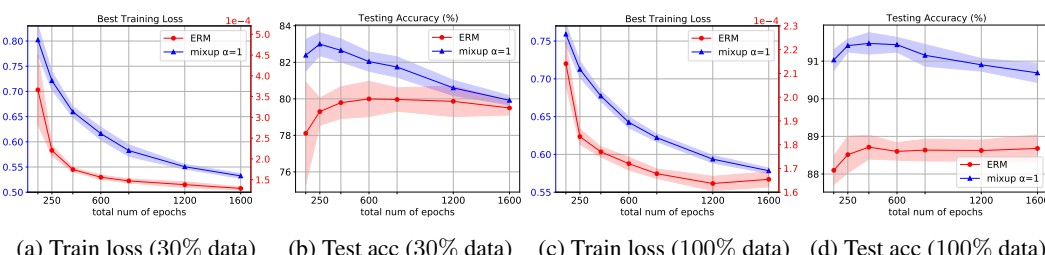

(a) Train loss (30% data)    (b) Test acc (30% data)    (c) Train loss (100% data)    (d) Test acc (100% data)

Figure 12: Results of the recorded training losses and testing accuracies of training VGG16 on CIFAR10 training set (30% data and 100% data) without data augmentation.

## B.2    RESULTS OF MEAN SQUARE ERROR LOSS WITHOUT DATA AUGMENTATION

We also perform Mixup training experiments using the mean square error (MSE) loss function on both CIFAR10 and SVHN datasets. Figure 13 illustrates that the U-shaped behavior observed in

previous experiments is also present when using the MSE loss function. To ensure optimal training, the learning rate is decreased by a factor of 10 at epoch 100 and 150.

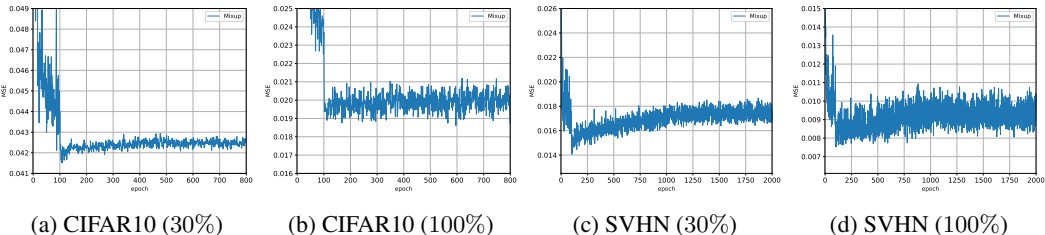

(a) CIFAR10 (30%)     (b) CIFAR10 (100%)     (c) SVHN (30%)     (d) SVHN (100%)

Figure 13: Dynamics of MSE during Mixup training.

### B.3   GRADIENT NORM VANISHES WHEN CHANGING MIXUP TO ERM

In Figure 8, we can observe that the gradient norm of Mixup training does not diminish at the end of training and can even explode to a very high value. In contrast, ERM results in a gradient norm of zero at the end of training. Figure 14 illustrates that when switching from Mixup training to ERM training after a certain period, the gradient norm will rapidly become zero. This phenomenon occurs because Mixup-trained neural networks have already learned the "clean patterns" and the original data does not provide any useful gradient signal. Therefore, this further supports the idea that the latter stage of Mixup training is primarily focused on memorizing noisy data.

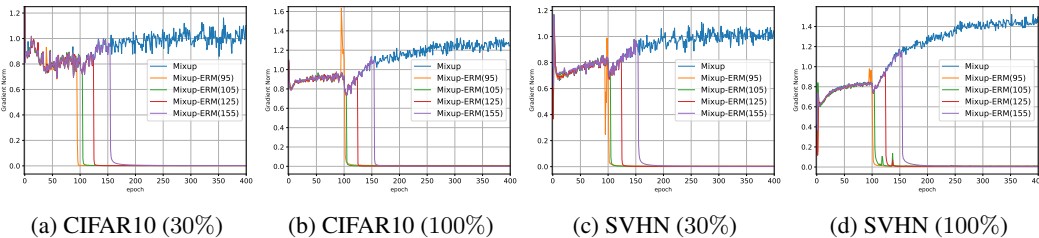

(a) CIFAR10 (30%)     (b) CIFAR10 (100%)     (c) SVHN (30%)     (d) SVHN (100%)

Figure 14: Dynamics of gradient norm when changing Mixup training to ERM training.

### B.4   VALIDATION RESULTS ON COVARIANCE-SHIFT DATASETS

Recall Figures 2 and 9, we have seen that as we increase the training epochs ($200 \rightarrow 400 \rightarrow 800$), the local minima on the loss landscape (measure by the real training data) of the Mixup-trained model gradually becomes sharper. To validate the regular pattern of the relationship between the minima flatness and the generalization behavior on the covariate-shift datasets, we have ran some of the Mixup-trained ResNet18 networks and tested their accuracies on CIFAR10.1 (Recht et al., 2018), CIFAR10.2 (Lu et al., 2020) and CIFAR10-C (Hendrycks & Dietterich, 2019) using Gaussian noise with severity 1 and 5 (denoted by CIFAR10-C-1 and CIFAR10-C-5). The results of the models pre-trained on 100% CIFAR10 are given in Figure 15, and the results of the models pre-trained on 30% CIFAR10 are given in Figure 16.

From the results, it is seen that with training epochs increase, the testing performance on the models on CIFAR10.1 and CIFAR10.2 decreases, taking a similar trend as our results in standard testing sets (i.e., the original CIFAR10 testing sets without covariate shift.) But on CIFAR10-C, this behaviour is not observed. In particular, the performance on CIFAR10-C-5 continues to improve over the training iterations. This seems to suggest that the flatness of empirical-risk loss landscape may impact generalization to covariate-shift datasets in more complex ways, possibly depending on the nature and structure of the covariate shift.

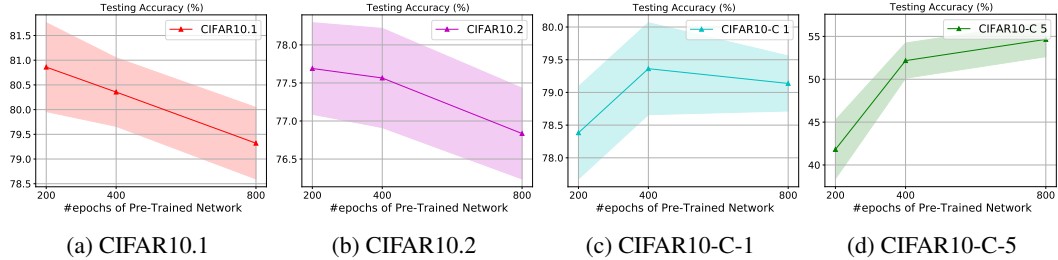

Figure 15: Models Pre-Trained on 100% CIFAR10 (without data augmentation)

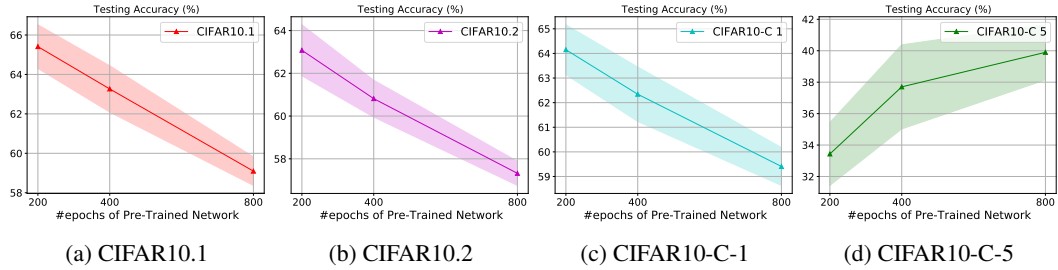

Figure 16: Models Pre-Trained on 30% CIFAR10 (without data augmentation)

## B.5 INVESTIGATION OF THE IMPACT OF REGMIXUP IN OVER-TRAINING

RegMixup is a variant algorithm of Mixup proposed by (Pinto et al., 2022). For each synthetic example $(\tilde{\mathbf{x}}, \tilde{\mathbf{y}})$ formulated by $(\mathbf{x}, \mathbf{y})$ and $(\mathbf{x}', \mathbf{y}')$, RegMixup minimizes the following loss

$$\ell_{\text{CE}}(\theta, \mathbf{x}, \mathbf{y}) + \eta \ell_{\text{CE}}(\theta, \tilde{\mathbf{x}}, \tilde{\mathbf{y}})$$

where $\ell_{\text{CE}}(\cdot)$ denotes the cross-entropy loss, and $\eta$ is non-negative. The authors show that RegMixup can improve generalization on both in-distribution and covariate-shift datasets, and that it can also improve the out-of-distribution robustness. To validate the performance of RegMixup in the over-training scenario, we have trained ResNet18 using RegMixup with a few different settings of $\eta$. The network is trained on CIFAR10 without data augmentation for up to in total 1200 epochs. The results are given in Figure 17.

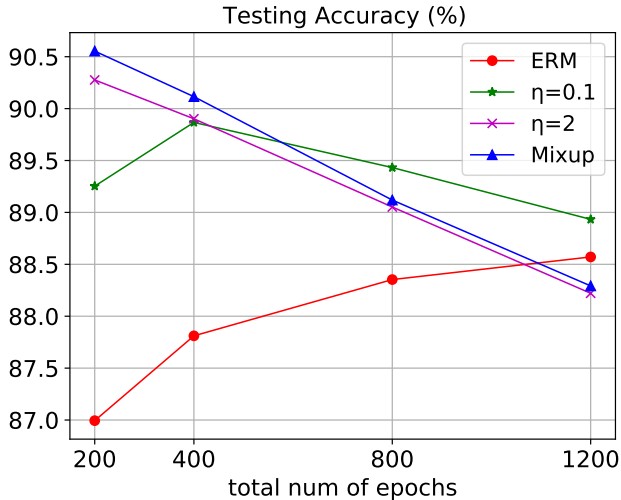

Figure 17: Results of Over-Training ResNet18 on CIFAR10 (without data augmentation) with Reg-Mixup. Green: $\eta = 0.1$; Purple: $\eta = 2$; Red: ERM; Blue: standard Mixup.

The results show that when $\eta = 2$, RegMixup performs nearly identically as standard Mixup in the over-training scenario. When $\eta = 0.1$, RegMixup postpones the presenting of the turning point, and in the large epochs it outperforms standard Mixup. However, the phenomenon that the generalization performance of the trained model degrades with over-training still exists.

## C EXPERIMENT SETTINGS FOR THE TEACHER-STUDENT TOY EXAMPLE

We set the dimension of the input feature as $d_0 = 10$. The teacher network consists of two layers with the activation function **Tanh**, and the hidden layer has a width of 5. Similarly, the student network is a two-layer neural network with **Tanh**, where we train only the second layer and keep the parameters in the first layer fixed. The hidden layer has a dimension of 100 (i.e. $d = 100$).

To determine the value of $\lambda$, we either draw from a **Beta(1,1)** distribution in each epoch or fix it to 0.5 in each epoch. We choose $n = 20$, which puts us in the overparameterized regime where $n < d$, and the underparameterized regime where $m \geq n^2 > d$. The learning rate is set to 0.1, and we use full-batch gradient descent to train the student network with MSE. Here, the term "full-batch" means that the batch size is equal to $n$, enabling us to compare the fixed $\lambda$ and random $\lambda$ methods fairly.

For additional information, please refer to our code.

### C.1 ABLATION STUDY: EFFECT OF FIXED MIXING COEFFICIENT

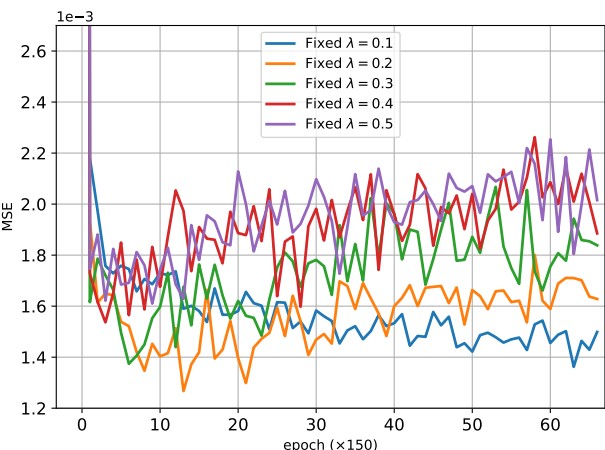

Figure 18: Results of the ablation study on $\lambda$.

In the teacher-student setting, we experiment with different fixed values of $\lambda$, and the results are presented in Figure 18. Of particular interest is the observation that as the noise level increases and $\lambda$ approaches 0.5, the turning point of testing error occurs earlier. This finding is consistent with our theoretical results.

## D OMITTED DEFINITIONS AND PROOFS

**Definition D.1** (Total Variation). *The total variation between two probability measures $P$ and $Q$ is* $\mathrm{TV}(P, Q) \triangleq \sup_E |P(E) - Q(E)|$, *where the supremum is over all measurable set $E$.*

**Lemma D.1** ((Levin & Peres, 2017, Proposition 4.2)). *Let $P$ and $Q$ be two probability distributions on $\mathcal{X}$. If $\mathcal{X}$ is countable, then*

$$\mathrm{TV}(P, Q) = \frac{1}{2} \sum_{x \in \mathcal{X}} |P(x) - Q(x)|.$$

**Lemma D.2** (Coupling Inequality ([Levin & Peres](), [2017](), Proposition 4.7))**.** *Given two random variables $X$ and $Y$ with probability distributions $P$ and $Q$, any coupling $\hat{P}$ of $P$, $Q$ satisfies*

$$\mathrm{TV}(P, Q) \leq \hat{P}(X \neq Y).$$

### D.1 PROOF OF LEMMA 3.1

*Proof.* We first prove the closed-form of the cross-entropy loss's lower bound. For any two discrete distributions $P$ and $Q$ defined on the same probability space $\mathcal{Y}$, the KL divergence of $P$ from $Q$ is defined as follows:

$$D_{\mathrm{KL}}(P\|Q) := \sum_{y \in \mathcal{Y}} P(y) \log\left(\frac{P(y)}{Q(y)}\right). \tag{4}$$

It is non-negative and it equals $0$ if and only if $P = Q$.

Let's denote the $i^{th}$ element in $f_\theta(x)$ by $f_\theta(x)_i$. By adapting the definition of the cross-entropy loss, we have:

$$
\begin{aligned}
\ell(\theta, (\mathbf{x}, \mathbf{y})) &= -\mathbf{y}^\mathsf{T} \log\left(f_\theta(\mathbf{x})\right) \\
&= -\sum_{i=1}^{K} y_i \log\left(f_\theta(\mathbf{x})_i\right) \\
&= -\sum_{i=1}^{K} y_i \log\left(\frac{f_\theta(\mathbf{x})_i}{y_i} y_i\right) \\
&= -\sum_{i=1}^{K} y_i \log\frac{f_\theta(\mathbf{x})_i}{y_i} - \sum_{i=1}^{C} y_i \log y_i \\
&= D_{\mathrm{KL}}\left(\mathbf{y}\|f_\theta(\mathbf{x})\right) + \mathcal{H}(\mathbf{y}) \\
&\geq \mathcal{H}(\mathbf{y}),
\end{aligned}
\tag{5}
$$

where the equality holds if and only if $f_\theta(\mathbf{x}) = \mathbf{y}$. Here $\mathcal{H}(\mathbf{y}) := \sum_{i=1}^{C} y_i \log y_i$ is the entropy of the discrete distribution $\mathbf{y}$. Particularly in ERM training, since $\mathbf{y}$ is one-hot, by definition its entropy is simply $0$. Therefore the lower bound of the empirical risk is given as follows.

$$\hat{R}_S(\theta) = \frac{1}{n} \sum_{i=1}^{n} \ell(\theta, (\mathbf{x}, \mathbf{y})) \geq 0 \tag{6}$$

The equality holds if $f_\theta(\mathbf{x}_i) = \mathbf{y}_i$ is true for each $i \in \{1, 2, \cdots, n\}$.

We then prove the lower bound of the expectation of empirical Mixup loss. From Eq. (5), the lower bound of the general Mixup loss for a given $\lambda$ is also given by:

$$
\begin{aligned}
\ell(\theta, (\tilde{\mathbf{x}}, \tilde{\mathbf{y}})) &\geq \mathcal{H}(\tilde{\mathbf{y}}) \\
&= -\sum_{i=1}^{C} y_i \log y_i \\
&= -\left(\lambda \log \lambda + (1 - \lambda) \log(1 - \lambda)\right).
\end{aligned}
\tag{7}
$$

if $(\tilde{\mathbf{x}}, \tilde{\mathbf{y}})$ is formulated via cross-class mixing. Recall the definition of the Mixup loss,

$$\hat{R}_{\widetilde{S}}(\theta, \alpha) = \mathop{\mathbb{E}}_{\lambda \sim Beta(\alpha, \alpha)} \frac{1}{n^2} \sum_{i=1}^{n} \sum_{j=1}^{n} \ell(\theta, (\tilde{\mathbf{x}}, \tilde{\mathbf{y}})), \tag{8}$$

we can exchange the computation of the expectation and the empirical average:

$$\hat{R}_{\widetilde{S}}(\theta, \alpha) = \frac{1}{n^2} \sum_{i=1}^{n} \sum_{j=1}^{n} \mathop{\mathbb{E}}_{\lambda \sim Beta(\alpha, \alpha)} \ell(\theta, (\tilde{\mathbf{x}}, \tilde{\mathbf{y}})) \tag{9}$$

Note that when $\alpha = 1$, $Beta(\alpha, \alpha)$ is simply the uniform distribution in the interval $[0, 1]$: $U(0, 1)$. Using the fact that the probability density of $U(0, 1)$ is constantly 1 in the interval $[0, 1]$, the lower bound of $\mathbb{E}_{\lambda \sim Beta(1,1)} \ell(\theta, (\tilde{\mathbf{x}}, \tilde{\mathbf{y}}))$ where $\mathbf{y} \neq \mathbf{y}'$ is given by:

$$
\begin{aligned}
\mathbb{E}_{\lambda \sim Beta(1,1)} \ell(\theta, (\tilde{\mathbf{x}}, \tilde{\mathbf{y}})) &\geq - \mathbb{E}_{\lambda \sim U(0,1)} \big( \lambda \log \lambda + (1 - \lambda) \log(1 - \lambda) \big) \\
&= - \int_0^1 \lambda \log \lambda + (1 - \lambda) \log(1 - \lambda) \, d\lambda \\
&= -2 \int_0^1 \lambda \log \lambda \, d\lambda \\
&= -2 \left( \log \lambda \int_0^1 \lambda \, d\lambda - \int_0^1 \frac{1}{\lambda} \left( \int_0^1 \lambda \, d\lambda \right) d\lambda \right) \\
&= -2 \left( \frac{\lambda^2 \log \lambda}{2} - \frac{\lambda^2}{4} \right) \Big|_0^1 \\
&= 0.5
\end{aligned}
\tag{10}
$$

Note that if the synthetic example is formulated via in-class mixing, the synthetic label is still one-hot, thus the lower bound of its general loss is 0. In a balanced $C$-class training set, with probability $\frac{1}{C}$ the in-class mixing occurs. Therefore, the lower bound of the overall Mixup loss is given as follows,

$$
\hat{R}_{\widetilde{S}}(\theta, \alpha = 1) \geq \frac{C - 1}{2C}
\tag{11}
$$

The equality holds if $f_\theta(\tilde{\mathbf{x}}) = \tilde{\mathbf{y}}$ is true for each synthetic example $(\tilde{\mathbf{x}}, \tilde{\mathbf{y}}) \in \widetilde{S}$. This completes the proof. $\square$

### D.2   PROOF OF THEOREM 5.1

*Proof.* By the coupling inequality i.e. Lemma D.2, we have

$$
\mathrm{TV}(P(\widetilde{Y}_{\mathrm{h}} | \widetilde{X}), P(\widetilde{Y}_{\mathrm{h}}^* | \widetilde{X})) \leq P(\widetilde{Y}_{\mathrm{h}} \neq \widetilde{Y}_{\mathrm{h}}^* | \widetilde{X}),
$$

Since $\mathrm{TV}(P(\widetilde{Y}_{\mathrm{h}} | \widetilde{X}), P(Y | X)) = \mathrm{TV}(P(\widetilde{Y}_{\mathrm{h}} | \widetilde{X}), P(\widetilde{Y}_{\mathrm{h}}^* | \widetilde{X}))$, then the first inequality is straightforward.

For the second inequality, by Lemma D.1, we have

$$
\begin{aligned}
\mathrm{TV}(P(\widetilde{Y}_{\mathrm{h}} | \widetilde{X}), P(\widetilde{Y}_{\mathrm{h}}^* | \widetilde{X})) &= \frac{1}{2} \sum_{j=1}^C \left| P(\widetilde{Y}^* = j | \widetilde{X}) - P(\widetilde{Y} = j | \widetilde{X}) \right| \\
&= \frac{1}{2} \sum_{j=1}^C \left| f_j(\widetilde{X}) - ((1 - \lambda) f_j(X) + \lambda f_j(X')) \right| \\
&\geq \sup_j \frac{1}{2} \left| f_j(\widetilde{X}) - ((1 - \lambda) f_j(X) + \lambda f_j(X')) \right|.
\end{aligned}
$$

This completes the proof. $\square$

### D.3   PROOF OF LEMMA 5.1

*Proof.* The ordinary differential equation of Eq. (2) (Newton's law of cooling) has the closed form solution

$$
\theta_t = \widetilde{\Phi}^\dagger \widetilde{\mathbf{Y}} + (\theta_0 - \widetilde{\Phi}^\dagger \widetilde{\mathbf{Y}}) e^{-\frac{\eta}{m} \widetilde{\Phi} \widetilde{\Phi}^T t}.
\tag{12}
$$

Recall that $\widetilde{\mathbf{Y}} = \widetilde{\mathbf{Y}}^* + \mathbf{Z}$,

$$
\begin{aligned}
\theta_t =& \widetilde{\Phi}^\dagger \left( \widetilde{\mathbf{Y}}^* + \mathbf{Z} \right) + (\theta_0 - \widetilde{\Phi}^\dagger \left( \widetilde{\mathbf{Y}}^* + \mathbf{Z} \right)) e^{-\frac{\eta}{m}\widetilde{\Phi}\widetilde{\Phi}^T t} \\
=& \widetilde{\Phi}^\dagger \widetilde{\mathbf{Y}}^* + \widetilde{\Phi}^\dagger \mathbf{Z} + \left( \theta_0 - \widetilde{\Phi}^\dagger \widetilde{\mathbf{Y}}^* \right) e^{-\frac{\eta}{m}\widetilde{\Phi}\widetilde{\Phi}^T t} - \widetilde{\Phi}^\dagger \mathbf{Z} e^{-\frac{\eta}{m}\widetilde{\Phi}\widetilde{\Phi}^T t} \\
=& \theta^* + (\theta_0 - \theta^*) e^{-\frac{\eta}{m}\widetilde{\Phi}\widetilde{\Phi}^T t} + (\mathbf{I}_d - e^{-\frac{\eta}{m}\widetilde{\Phi}\widetilde{\Phi}^T t}) \theta^{\text{noise}},
\end{aligned}
$$

which concludes the proof. $\qquad\square$

## D.4 PROOF OF THEOREM 5.2

*Proof.* We first notice that

$$
\begin{aligned}
R_t =& \mathbb{E}_{\theta_t, X, Y} \left|\left| \theta_t^T \phi(X) - Y \right|\right|_2^2 \\
=& \mathbb{E}_{\theta_t, X, Y} \left|\left| \theta_t^T \phi(X) - \theta^{*T}\phi(X) + \theta^{*T}\phi(X) - Y \right|\right|_2^2 \\
=& \mathbb{E}_{\theta_t, X} \left|\left| \theta_t^T \phi(X) - \theta^{*T}\phi(X) \right|\right|_2^2 + \mathbb{E}_{X, Y} \left|\left| \theta^{*T}\phi(X) - Y \right|\right|_2^2 + 2\mathbb{E}_{\theta_t, X, Y} \langle \theta_t^T \phi(X) - \theta^{*T}\phi(X), \theta^{*T}\phi(X) - Y \rangle \\
\leq& \mathbb{E}_X \left|\left| \phi(X) \right|\right|_2^2 \mathbb{E}_{\theta_t} \left|\left| \theta_t^T - \theta^{*T} \right|\right|_2^2 + R^* + 2\sqrt{\mathbb{E}_{\theta_t, X} \left|\left| \theta_t^T \phi(X) - \theta^{*T}\phi(X) \right|\right|_2^2 \mathbb{E}_{X, Y} \left|\left| \theta^{*T}\phi(X) - Y \right|\right|_2^2} \\
\leq& \frac{C_1}{2} \mathbb{E}_{\theta_t} \left|\left| \theta_t^T - \theta^{*T} \right|\right|_2^2 + R^* + 2\sqrt{\frac{C_1 R^*}{2} \mathbb{E}_{\theta_t} \left|\left| \theta_t^T - \theta^{*T} \right|\right|_2^2}, \qquad (13)
\end{aligned}
$$

where the first inequality is by the Cauchy–Schwarz inequality and the second inequality is by the assumption.

Recall Eq. (3),

$$
\theta_t - \theta^* = (\theta_0 - \theta^*) e^{-\frac{\eta}{m}\widetilde{\Phi}\widetilde{\Phi}^T t} + (\mathbf{I}_d - e^{-\frac{\eta}{m}\widetilde{\Phi}\widetilde{\Phi}^T t}) \widetilde{\Phi}^\dagger \mathbf{Z}.
$$

By eigen-decomposition we have

$$
\frac{1}{m}\widetilde{\Phi}\widetilde{\Phi}^T = V\Lambda V^T = \sum_{k=1}^d \mu_k v_k v_k^T,
$$

where $\{v_k\}_{k=1}^d$ are orthonormal eigenvectors and $\{\mu_k\}_{k=1}^d$ are corresponding eigenvectors.

Then, for each dimension $k$,

$$
(\theta_{t,k} - \theta_k^*)^2 \leq 2(\theta_{0,k} - \theta_k^*)^2 e^{-2\eta\mu_k t} + 2(1 - e^{-\eta\mu_k t})^2 \frac{mC_2}{m\mu_k},
$$

Taking expectation over $\theta_0$ for both side, we have

$$
\mathbb{E}_{\theta_0} (\theta_{t,k} - \theta_k^*)^2 \leq 2(\xi_k^2 + \theta_k^{*2}) e^{-2\eta\mu_k t} + 2(1 - e^{-\eta\mu_k t})^2 \frac{C_2}{\mu_k}. \qquad (14)
$$

Notich that the RHS in Eq. 14 first monotonically decreases and then monotonically increases, so the maximum value of RHS is achieved either at $t = 0$ or $t \to \infty$. That is,

$$
\mathbb{E}_{\theta_0} \left|\left| \theta_t^T - \theta^{*T} \right|\right|_2^2 \leq \sum_{k=1}^d 2\max\{\xi_k^2 + \theta_k^{*2}, \frac{C_2}{\mu_k}\}. \qquad (15)
$$

Plugging Eq. 14 and Eq. 15 into Eq. 13, we have

$$
\begin{aligned}
R_t \leq& \frac{C_1}{2} \mathbb{E}_{\theta_t} \left|\left| \theta_t^T - \theta^{*T} \right|\right|_2^2 + R^* + 2\sqrt{\frac{C_1 R^*}{2} \mathbb{E}_{\theta_t} \left|\left| \theta_t^T - \theta^{*T} \right|\right|_2^2} \\
\leq& R^* + C_1 \sum_{k=1}^d (\xi_k^2 + \theta_k^{*2}) e^{-2\eta\mu_k t} + (1 - e^{-\eta\mu_k t})^2 \frac{C_2}{\mu_k} + 2\sqrt{C_1 R^* \zeta},
\end{aligned}
$$

where $\zeta = \sum_{k=1}^d \max\{\xi_k^2 + \theta_k^{*2}, \frac{C_2}{\mu_k}\}$. This concludes the proof. $\qquad\square$

