# OpenReview forum: "Over-Training with Mixup May Hurt Generalization"
_ICLR.cc/2023/Conference — ICLR 2023 poster_

### Official Review · Reviewer_gGCe · 2022-10-22

**Confidence:** 4
**Correctness:** 3
**Technical Novelty And Significance:** 3
**Empirical Novelty And Significance:** 4
**Recommendation:** 5

**Clarity, Quality, Novelty And Reproducibility:**

The writing is clear, and the paper is structured logically. Maybe the figures could be positioned a bit better, and some visual diversity could be used to help parse them (e.g. aggregating some plots where reasonable and using colours better).
The work is original enough and interesting. The experiments seem to be easily reproducible by others given Mixup is a well-established technique.

**Strength And Weaknesses:**

Strengths:
- clearly written, easy to follow, good references
- the phenomenon is novel, although overtraining is not typically a problem in large-scale experimental settings
- most explanations provided are plausible and supported either by theory or experiments

Weaknesses:
- Q1: The experiments mostly use ResNets. Does this phenomenon emerge also for other types of architectures? The way overparametrisation plays a role might interact differently with different inductive biases. Could you try showing some results for other convolutional architectures (e.g. ConvNeXt, PyramidNet), some sequential tasks (e.g. using LSTMs)  or Transformers fine-tuning?
- Q2: The authors refer to the conjecture that flatter minima imply better generalization: for CIFAR-10 and CIFAR-100 covariate-shift datasets are available (e.g. CIFAR-10/100-C, CIFAR-10.1 and CIFAR-10.2). Would it be possible for them to validate their argument about the flatness inducing better generalization on these datasets by running their models and evaluating the accuracy on these datasets? Indeed, whether the Mixup loss is flat or sharp should not bee too relevant, what probably should matter is whether in the loss-landscape of the ERM cross-entropy loss the configuration reached by the optimisation algorithm is flat or not, and this will influence the actual generalization performance.
- Q3: I understand the authors probably switched off augmentations to disentangle the behaviour of Mixup from that of augmentations. However, the authors claim augmentations would further exasperate the phenomenon observed. Could they provide empirical evidence for this?
- Q4: It is not clear to me how the teacher-student experiment should be relevant. What is it supposed to tell? Maybe it's just a matter of writing or my own misunderstanding. Could you please elaborate further?
- Q5: Mixup is a technique used in many SOTA training procedures. The authors seem to suggest it could be useful to switch it off after a few epochs. This creates the issue of having to determine when it should be switched off: presumably, choosing different epochs will produce very different results. Is there a procedure the authors could suggest to do it? Additionally, the effectiveness of this procedure should be validated on more extensive experimental evidence (a few more architectures, and if possible larger datasets (e.g. ImageNet)).
- Q6: The authors of [1] suggest using the Mixup loss as a regulariser of normal cross-entropy (i.e. the loss is CE(clean samples) + MixupLoss(interpolatedSamples)) in order to fix out-of-distribution detection issues given by the label-smoothing effect of Mixup induced by the randomness of the interpolation parameter.  Given the argument you make about the impact of the label noise and you suggest a fix by switching off Mixup, could you please either empirically or theoretically discuss whether using Mixup as a regularizer of cross-entropy could be a fix to the problem? Would the impact of the noise signal be alleviated or cancelled in this case?
- Q7: Overtraining is not really a problem for large-scale classification (since the resources requirements are already extremely high that no one would waste them overtraining), however your observations can be helpful in case practitioners observe failure cases of Mixup on smaller datasets. Instead of subsampling larger datasets, could you suggest real-world datasets in which overtraining might happen unintentionally and hence make practitioners actually observe this phenomenon?
- Q8: Since Mixup already requires cross-validation (given the need of cross-validating the hyperparameter of the Beta distribution), what would prevent practitioners to just use Early Stopping or some other strategy to avoid over-training? Would Early Stopping outperform ERM training?
[1] RegMixup: Mixup as a Regularizer Can Surprisingly Improve Accuracy and Out Distribution Robustness; Francesco Pinto, Harry Yang, Ser-Nam Lim, Philip H.S. Torr, Puneet K. Dokania




**Summary Of The Paper:**

The authors point out a previously unknown behaviour of Mixup when overtraining: when the number of training epochs is extremely high, the Mixup performance is inferior to the one of ERM. The authors provide sufficient empirical evidence to show this phenomenon generalizes at least across datasets, and provide an explanation in terms of the labeling noise Mixup induces. They also provide a simplilfied theoretical explanation on a random feature model that exhibits a similar behaviour. As a fix to the problem, they suggest switching off mixup after a few epochs, which is empirically interesting, although further experiments would be required in order to understand its usefulness for more complex setups.

**Summary Of The Review:**

The paper is overall interesting, although its practical usefulness is not particularly emphasized. The idea is simple and clear, the explanations are plausible. If the authors could convincingly address the weaknesses I pointed out, I'd be happy to rise the score.

---

> ### Author Response · Authors · 2022-11-17
> **To Reviewer gGCe**
>
> We thank you sincerely for your comments to our paper. Our responses follow.
>
> >- The experiments mostly use ResNets. Does this phenomenon emerge also for other types of architectures? The way overparametrisation plays a role might interact differently with different inductive biases. Could you try showing some results for other convolutional architectures (e.g. ConvNeXt, PyramidNet), some sequential tasks (e.g. using LSTMs) or Transformers fine-tuning?
>
> We conducted the over-training experiments using VGG16 on CIFAR10 (100% training data and 30% training data respectively) without data augmentation. The results are added in Appendix B.1. In these experiments, we observe similar phenomenon as in those with ResNet18 or ResNet34. This should suggest convincingly that such a behavior of Mixup is not exclusive to the residual networks.
>
> >- The authors refer to the conjecture that flatter minima imply better generalization: for CIFAR-10 and CIFAR-100 covariate-shift datasets are available (e.g. CIFAR-10/100-C, CIFAR-10.1 and CIFAR-10.2). Would it be possible for them to validate their argument about the flatness inducing better generalization on these datasets by running their models and evaluating the accuracy on these datasets? Indeed, whether the Mixup loss is flat or sharp should not bee too relevant, what probably should matter is whether in the loss-landscape of the ERM cross-entropy loss the configuration reached by the optimisation algorithm is flat or not, and this will influence the actual generalization performance.
>
> We have performed additional experiments on the covariate-shift datasets: CIFAR10.1, CIFAR10.2, CIFAR10-C with Gaussian noise of severity levels 1 and 5 (denoted by CIFAR10-C-1, CIFAR10-C-5). Previously trained models on 30% and 100% CIFAR10 are tested on these datasets. The results are presented in Appendix B.5.
>
> From the results, it is seen that with training epochs increases, the testing performance on the models on CIFAR10.1 and CIFAR10.2 decreases, taking a similar trend as our results in standard testing sets (i.e., the original CIFAR10 testing sets without covariate shift.) But on CIFAR10-C, this behaviour is not observed. In particular, the performance on CIFAR10-C-5 continues to improve over training iterations. This seems to suggest that the flatness of empirical-risk loss landscape may impact generalization to covariate-shift datasets in more complex ways, possibly depending on the nature and structure of the covariate shift.
>
> Although it is of great research interest to study the impact of Mixup training on covariate-shift datasets and we will include these results in the paper, the subject appears only marginally related to this work.
>
> >- I understand the authors probably switched off augmentations to disentangle the behaviour of Mixup from that of augmentations. However, the authors claim augmentations would further exasperate the phenomenon observed. Could they provide empirical evidence for this?
>
> First, we wish to clarify that the submitted paper made no claim regarding the effect of data augmentation in this context. In the revision (Appendix B.2), we added a discussion about data augmentation and presented additional experimental results. Please refer to "Response to All Reviewers".
>
> >- It is not clear to me how the teacher-student experiment should be relevant. What is it supposed to tell? Maybe it's just a matter of writing or my own misunderstanding. Could you please elaborate further?
>
> We wish to note that the teacher-student experiment is not related to the knowledge distillation setting, in which the term commonly occurs. In our setting, the teacher model simply serves as a ground-truth, using which simulated data can be generated. The student model is assumed to be ignorant about the ground truth and then tries to learn it using these simulated data via MixUp training. This allows us to accurately evaluate population risk of learned student model and validate the theoretical results in Section 5.2.
>
> >- Mixup is a technique used in many SOTA training procedures. The authors seem to suggest it could be useful to switch it off after a few epochs. This creates the issue of having to determine when it should be switched off: presumably, choosing different epochs will produce very different results. Is there a procedure the authors could suggest to do it? Additionally, the effectiveness of this procedure should be validated on more extensive experimental evidence (a few more architectures, and if possible larger datasets (e.g. ImageNet)).
>
> We wish to note that switching off MixUp mainly serves as experimental validation for our theoretical analysis. It is not meant to create a SOTA regularization scheme. Please also refer to "Response to All Reviewers".

---

> > ### Author Response · Authors · 2022-11-17
> > **To Reviewer gGCe (cont.)**
> >
> > >- The authors of [1] suggest using the Mixup loss as a regulariser of normal cross-entropy (i.e. the loss is CE(clean samples) + MixupLoss(interpolatedSamples)) in order to fix out-of-distribution detection issues given by the label-smoothing effect of Mixup induced by the randomness of the interpolation parameter. Given the argument you make about the impact of the label noise and you suggest a fix by switching off Mixup, could you please either empirically or theoretically discuss whether using Mixup as a regularizer of cross-entropy could be a fix to the problem? Would the impact of the noise signal be alleviated or cancelled in this case?
> >  [1] RegMixup: Mixup as a Regularizer Can Surprisingly Improve Accuracy and Out Distribution Robustness; Francesco Pinto, Harry Yang, Ser-Nam Lim, Philip H.S. Torr, Puneet K. Dokania
> >
> > We have performed the over-training experiments using RegMixup on ResNet18. The network is trained on 100% CIFAR10 without data augmentation for upt to in total 1200 epochs. We set the parameter $\eta$ of RegMixup as 0.1 and 2 respectively, and the result is given in Appendix B.6.
> >
> > The results show that when $\eta=2$, RegMixup performs nearly identically as standard Mixup in the over-training scenario. When $\eta=0.1$, RegMixup postpones the presenting of the turning point, and in the large epochs it outperforms standard Mixup. However, the phenomenon that the generalization performance of the trained model degrades with over-training still exists.
> >
> > >- Overtraining is not really a problem for large-scale classification (since the resources requirements are already extremely high that no one would waste them overtraining), however your observations can be helpful in case practitioners observe failure cases of Mixup on smaller datasets. Instead of subsampling larger datasets, could you suggest real-world datasets in which overtraining might happen unintentionally and hence make practitioners actually observe this phenomenon?
> >
> > In practice, when the algorithm designer is designing new Mixup-like schemes, if the intended scheme aggressively modifies the input examples, it may have a similar, or even worse effect, as using $\lambda=0.5$ in Mixup. In this case, the designer should be cautious about the possibility of introducing strong label noise and carefully inspect the training process to avoid over-training. Please also refer to "Response to All Reviewers".
> >
> > >- Since Mixup already requires cross-validation (given the need of cross-validating the hyperparameter of the Beta distribution), what would prevent practitioners to just use Early Stopping or some other strategy to avoid over-training? Would Early Stopping outperform ERM training?
> >
> > First from our observations, Mixup with early stop at appropriate training iterations will have a better performance than ERM. The question is often what is the appropriate iteration number at which training should be stopped. When a validation set of abundant size is available, cross-validation on the validation set is always a good solution. One take-away from this work is that under Mixup training, cross-validation is still necessary. It is dangerous to assume that Mixup won't overfit or that the longer training the better.

---

### Official Review · Reviewer_Zo8A · 2022-10-25

**Confidence:** 4
**Correctness:** 3
**Technical Novelty And Significance:** 3
**Empirical Novelty And Significance:** 3
**Recommendation:** 5

**Clarity, Quality, Novelty And Reproducibility:**

The paper has good clarity, makes a few novel contributions, and the authors have provided code for reproducibility.
Concerns regarding the quality of the paper have been discussed in the "weakness" section.

**Strength And Weaknesses:**

Strength:
- the paper studies an interesting phenomenon concerning a popular training loss (or data augmentation method), which has a wide audience base;
- the paper provides both empirical evidence and theoretical arguments to support its hypothesis;
- the paper is well-written and easy to follow.

Weakness:
- On the novelty and validity of the reported phenomenon. With some search I found a paper [1] reporting test accuracies during extended training with ERM or mixup, on benchmark datasets such as CIFAR-10 and CIFAR-100. In [1, Table 13 and 14] only one out of four settings (i.e. CIFAR-100 + RX-50) shows performance degradation with extended mixup training. One significant difference of the current work is that it turns off (other) data augmentation. Could this be the reason why the reported results seem contradictory? And if so, maybe it is proper to state that the observation may only be valid without data augmentation, and contrast with the previous report.

[1] AutoMix: Unveiling the Power of Mixup for Stronger Classifiers. ECCV 2022.

- On the interpretation of the theoretical analysis. If I understand correctly, the gradient flow characterization (eq. (3), Lemma 5.1) is basically the coordinates of $\theta_t$ as a parametric curve. While $\theta_t \neq \theta^*$, it does not imply that $||\theta_t - \theta^*||$ or $R_t$ is a U-shaped function. Theorem 5.2 is an upper bound result on $R_t - R^*$. Without a matching lower bound, or a direct calculation of $\frac{d R_t}{dt}$, or showing $R_t$ is smaller for some $t$ vs. $t \to \infty$, one still cannot justify that $R_t$ will **actually** increase as $t$ goes to infinity.

There are two additional points regarding the link between the authors' theory and observation that I would raise for discussion:

1. At the end of Sec. 5, the authors claim that their Theorem 5.2 "explains why reducing the learning rate will make the population risk again decrease in a certain interval". However, as I understand, in the gradient flow ODE interpretation (eq. (3), Lemma 5.1), changing $\eta$ simply corresponds to reparametrize the curve, (e.g. reducing $\eta$ by half corresponds to running $t$ 2x slower), hence a stretching or squeezing of the $R_t$ curve along the x-axis. Therefore it only scales $\frac{d R_t}{dt}$ but does not change its sign. Could you explain your reasoning and correct me if I am wrong?

2. At the end of Sec. 6.1, the authors claim that the minimum of the MSE loss for a fixed $\lambda=0.5$ come earlier in training due to increased noise level, which is consistent with their theory. However, as drawing $\lambda$ from a distribution introduces extra noise to the training, a better ablation study would be to compare the effects of different fixed $\lambda$, e.g. $\\lambda \\in \\{0.1, 0.2, 0.3, 0.4, 0.5\\}$.

**Summary Of The Paper:**

In this paper, the authors studied why training with the mixup loss for extra epochs often degrades test classification accuracy. They argued, through empirical observation, theoretical analysis and synthetic experiments, that the mixup loss introduces label noise, and a model may overfit to the label noise during extended training, hence increasing the generalization gap.

This paper makes several contributions:
1. it reports an interesting phenomenon in training classifiers and replicates it on multiple datasets;
2. it shows that extended training with the mixup loss increases the sharpness of the loss landscape;
3. it provides a possible explanation to the phenomenon by analyzing a regression problem;
4. it creates a synthetic regression problem which validates the theoretical analysis;
5. it suggests a practical solution to alleviate the observed problem.

**Summary Of The Review:**

To my best knowledge, I believe the paper studies an interesting and potentially important question; both the experiments and the theoretical analysis are sound per se; however, I have concerns regarding the novelty and scope of validity of the reported phenomenon, as well as whether the theoretical analysis actually justifies the authors' hypothesis.

---

> ### Author Response · Authors · 2022-11-17
> **To Reviewer Zo8A**
>
> We thank you sincerely for your comments to our paper. Our responses follow.
>
> >- On the novelty and validity of the reported phenomenon. With some search I found a paper [1] reporting test accuracies during extended training with ERM or mixup, on benchmark datasets such as CIFAR-10 and CIFAR-100. In [1, Table 13 and 14] only one out of four settings (i.e. CIFAR-100 + RX-50) shows performance degradation with extended mixup training. One significant difference of the current work is that it turns off (other) data augmentation. Could this be the reason why the reported results seem contradictory? And if so, maybe it is proper to state that the observation may only be valid without data augmentation, and contrast with the previous report.
> >- [1] AutoMix: Unveiling the Power of Mixup for Stronger Classifiers. ECCV 2022.
>
> Please refer to "Response to All Reviewers" for our discussion of data augmentation, where we have argued with experimental evidence that even with data augmentation the presented phenomenon may still exist. It is true that in [1], performance degradation is not observed for all datasets. We believe that this is because the training in those cases has not entered the over-training regime. In [1], the maximum number of training epochs is taken as 1200. In our experiments (Appendix B.2), on 10% CIFAR10 and 10% CIFAR100 datasets with data augmentation, we run training for up to 7000 epochs in order to show the U-shaped error curve. On the original CIFAR10 and CIFAR100 with data augmentation, we expect an even larger number of training iterations before observing the U-shaped curve.
>
> >- On the interpretation of the theoretical analysis. If I understand correctly, the gradient flow characterization (eq. (3), Lemma 5.1) is basically the coordinates of $\theta_t$ as a parametric curve. While $\theta_t\neq\theta^*$, it does not imply that $\Vert \theta _ t-\theta^* \Vert$ or $R_t$ is a U-shaped function. Theorem 5.2 is an upper bound result on $R_t-R^*$. Without a matching lower bound, or a direct calculation of $\frac{dR_t}{dt}$, or showing $R_t$ is smaller for some $t$ vs. $t\rightarrow\infty$, one still cannot justify that $R_t$ will **actually** increase as $t$ goes to infinity.
>
> It is indeed the case that Theorem 5.2 only provides an upper bound, and the U-shaped upper bound needs not to guarantee a U-shaped true error curve. However, we wish to argue that the U-shaped error curve is more likely to arise based on the results of Lemma 5.1. In that lemma, $\theta^*$ is the true solution for the regression problem, and $\theta^{\rm noise}$ is created by the Mixup-induced label noise. From Equation (3), it is possible to see that the trajectory of $\theta^t$ in early training stage is dominated by the first term, which pulls $\theta^t$ towards $\theta^*$. In the later training stage, the second term of Equation (3) dominates the trajectory. Then $\theta^t$ is pulled towards $\theta^*+\theta^{\rm noise}$. When  $\theta^{\rm noise}$ is relatively large (corresponding to strong label noise induced by Mixup), the true solution $\theta^*$ is sufficiently apart from the spurious solution $\theta^*+ \theta^{\rm noise}$. Hence the true error should first decay (as $\theta^t$ moves to $\theta^*$) and then increase (as $\theta^t$ moves to $\theta^*+\theta^{\rm noise}$). Of course, to precisely argue the existence of the U-shaped curve, one needs to carefully inspect the relative locations among the initial position $\theta_0$, the true solution $\theta^*$ and the spurious solution $\theta^*+\theta^{\rm noise}$. In particular, when $\theta^*$ has small norm, making $\theta^*+\theta^{\rm noise}$ very close to $\theta^*$, the U-shaped error curve may not occur.

---

> > ### Author Response · Authors · 2022-11-17
> > **To Reviewer Zo8A (cont.)**
> >
> > >- At the end of Sec. 5, the authors claim that their Theorem 5.2 "explains why reducing the learning rate will make the population risk again decrease in a certain interval". However, as I understand, in the gradient flow ODE interpretation (eq. (3), Lemma 5.1), changing $\eta$ simply corresponds to reparameterize the curve, (e.g. reducing $\eta$ by half corresponds to running $t$ 2x slower), hence a stretching or squeezing of the $R_t$ curve along the x-axis. Therefore it only scales $\frac{dR_t}{dt}$ but does not change its sign. Could you explain your reasoning and correct me if I am wrong?
> >
> > This confusion arises from the sloppiness in our presentation. We have revised the writing for an improved clarity, which we also explain here.
> >
> > The consideration here is a multi-stage training set up, where the learning rate is reduced after certain number of epochs. Suppose that with the initial learning rate, at epoch T, the test error has dropped to the bottom of the U-curve. If the learning rate is decreased at this point, then the U-shape curve corresponding to the new learning rate may have a lower minimum error and its bottom shifted to the right. In this case, the new learning rate allows the testing error to follow the new U-shaped curve and hence further decay.
> >
> > >- At the end of Sec. 6.1, the authors claim that the minimum of the MSE loss for a fixed $\lambda=0.5$ come earlier in training due to increased noise level, which is consistent with their theory. However, as drawing $\lambda$ from a distribution introduces extra noise to the training, a better ablation study would be to compare the effects of different fixed $\lambda$, e.g. $\lambda \in${$0.1,0.2,0.3,0.4,0.5$}.
> >
> > Thanks for the suggestion. We have added the ablation study on the effect of fixed $\lambda$ in the revision (see Figure 16 in Appendix C.1), using the suggested $\lambda$ values. It can be seen that the closer is $\lambda$ to $0.5$, the earlier will the minimum of the MSE loss arrive.

---

### Official Review · Reviewer_9xtA · 2022-10-30

**Confidence:** 3
**Correctness:** 3
**Technical Novelty And Significance:** 3
**Empirical Novelty And Significance:** 3
**Recommendation:** 8

**Clarity, Quality, Novelty And Reproducibility:**

The paper is generally quite well-written. However, there are a few issues which should be addressed:

1. The presentation of the theorem (particularly Thm 5.2) could be made clearer to have the takeaways be easier to extract. There also seems to be an error in the statement, I believe there should be a sum over all k in front of the C_2/\mu_k term. Due to all the \mu_k's and \sigma_k's floating around, it is not as easy to understand how the parameters interact. Perhaps having a corollary for some simple (e.g. isotropic?) settings would be useful.
2. The text in the figures is too small to read.



**Strength And Weaknesses:**

Strengths:

1. The paper identifies an interesting empirical phenomenon, and the findings appear to be quite robust to different setups.
2. The theory is clean, and provides a clear explanation which fits the intuition.
3. The paper relates the observations to a number of recent empirical phenomenon identified in deep learning, such as sharp/flat minima and the gradient norms not converging to zero. This places the paper well in the broader literature, and it could contribute to a growing understanding of neural network training dynamics.

Weaknesses:

1. The finding might have slightly limited practical applicability since the #epochs at which overfitting starts to happen is quite large, and early-stopping is often employed when the model starts to overfit. The main takeaway is probably the more conceptual one of how mixup training behaves.

**Summary Of The Paper:**

The paper aims to understand Mixup training, and specifically the training dynamics if mixup is continued to run for much longer than is usual. The paper shows an interesting empirical phenomenon, that overtraining with mixup leads to a U-shaped curve for training error, i.e. training error increases after a point (and also becomes worse than the error of the ERM solution). To demonstrate this, the paper considers ResNet models trained on CIFAR10 and SVHN datasets.

The paper then provides an intuitive explanation for this phenomenon for the simple regression setting on Gaussian data. Though the setup is simplistic, it brings out the same phenomenon as in the more realistic experiments. The key idea behind why Mixup overfits when overtrained is intuitive too: if the ground truth function f is non-linear, then the Mixup labels do not necessarily correspond to the labels assigned by the ground-truth function (because Mixup does a linear interpolation to assign the labels of the Mixed-up datapoints which would be inconsistent with the labels if f is non-linear). Therefore, Mixup introduces label noise, and if overtrained the model will fit the noise and not generalize as well.



**Summary Of The Review:**

In summary, I think this is an interesting paper and provides valuable insight into a very popular training strategy (Mixup). While it may not directly lead to better training, I think it does provide useful insight into neural network training dynamics more broadly. This aspect is further strengthened by the papers analysis of the overtraining phenomenon in relation to other empirically observed phenomenon in deep learning.

---

> ### Author Response · Authors · 2022-11-17
> **To Reviewer 9xtA**
>
> We thank you sincerely for your comments to our paper. Our responses follow.
>
> >- The finding might have slightly limited practical applicability since the #epochs at which overfitting starts to happen is quite large, and early-stopping is often employed when the model starts to overfit. The main takeaway is probably the more conceptual one of how Mixup training behaves.
>
> Please refer to "Response to All Reviewers".
>
> >- The presentation of the theorem (particularly Thm 5.2) could be made clearer to have the takeaways be easier to extract. There also seems to be an error in the statement, I believe there should be a sum over all $k$ in front of the $C_2/\mu_k$ term. Due to all the $\mu_k$'s and $\sigma_k$'s floating around, it is not as easy to understand how the parameters interact. Perhaps having a corollary for some simple (e.g. isotropic?) settings would be useful.
>
> Thank you for pointing to the error in Theorem 5.2. Indeed we missed a pair of brackets. The error has been fixed in the revised version.
> Following your suggestion, we added the Remark 5.5 in the revision to simplify the upper bound, so as to show how various parameters interact.
>
> >- The text in the figures is too small to read.
>
> We have enlarged the fontsizes of the text in the figures.

---

### Official Review · Reviewer_VPXk · 2022-10-31

**Confidence:** 3
**Clarity, Quality, Novelty And Reproducibility:** Some ambiguity remains, but the resul…
**Correctness:** 3
**Technical Novelty And Significance:** 2
**Empirical Novelty And Significance:** 2
**Recommendation:** 6

**Strength And Weaknesses:**

This paper provides beneficial practical insights. Many practitioners will be interested in this issue related to the mixup and overtraining. The authors' contribution is to raise the issue based on experimental observations and present a simple theoretical framework.

The theoretical result is straightforward and may not be accurate in some points. Furthermore, I felt that the theoretical validation between the analysis of classification (Section 5.1) and the analysis of regression (Section 5.2) is not consistent. Therefore, there is much room for improvement.

However, the problem setting and the approach are interesting. Although I think that the paper's proposed framework is not complete, I am inclined to accept it because I appreciate the challenging attempts.

**Summary Of The Paper:**

This paper investigates the phenomenon that the predictive performance of a prediction function degrades when overtraining under mixup data augmentation. The authors refer to the loss landscape as the U-shaped curve, where the prediction performance on the test data degrades when the number of epochs is increased. After observing the phenomenon experimentally, the authors present a theory that explains the phenomenon. The authors consider both classification and regression problems, although they focus primarily on classification problems.

**Summary Of The Review:**

### Major comments:
The problem setting and the theoretical framework proposed in this paper are very simple. While the theoretical analysis could be improved, the problem posed and the experimental observations are intriguing.

The reason why I feel that the theoretical analysis is incomplete is that it is hard to find the consistency among some of the results presented. For example, the connection between Theorem 5.1 and Theorem 5.2 is not clear. Overall, the results seem to be pieced together with little relevance. Furthermore, there are some ambiguous descriptions and undefined terms.

Although I am dissatisfied with some of such details, I will not vote for rejection because it reports findings that are useful for practical use.

- On page 3: In Lemma 3.1, what is $\theta$ or what is a class of $\theta$? For example, can the authors write, "Then, for all $\theta \in ???$"?
- On page 3: In Lemma 3.1, $\mathbb{E}_{\lambda}\hat{R}(\theta)$ is a random variable regarding the randomness of $\lambda$? That is,  does the statement hold with probability one?
- The figures are hard to read because the characters are very small. It is not kind to people who read the printed paper.
- While the y-axis of the experimental results for test data in Figure 1 is test error, it is test accuracy in Figure 2. I read a printed paper, so I could not notice the change in the y-axis and was confused.
- On page 5: In Theorem 5.1, does the bound hold with probability one?
- On page 6: Is the definition that "For example, if $f$ is strongly convex with some parameter..." applied for all $x$?
- I omit other details but feel that the definition and statement in the theoretical analysis are not rigorous. I would like the authors to clarify them in more detail.

### Minor comments:
- On page 3: Can we interpreted $\mathcal{Y}$ as being $\mathcal{Y} \in \mathcal{P}(\mathcal{Y})$.
- On page 3: In Lemma 3.1, what is the definition of $\mathbb{E}_{\lambda}$?

---

> ### Author Response · Authors · 2022-11-17
> **To Reviewer VPXk**
>
> We thank you sincerely for your comments to our paper. Our responses follow.
>
> >The reason why I feel that the theoretical analysis is incomplete is that it is hard to find the consistency among some of the results presented. For example, the connection between Theorem 5.1 and Theorem 5.2 is not clear. Overall, the results seem to be pieced together with little relevance. Furthermore, there are some ambiguous descriptions and undefined terms.
>
> Regarding Theorems 5.1 and 5.2, we agree that they are not perfectly connected. Specifically Theorem 5.1 suggests that for any classification problem, Mixup training induces label noise, whereas Theorem 5.2 (and the entire section 5.2) analyzes a regression problem and explain the label noise may result in the U-shape learning curve. The choice of the regression problem in this analysis is due to the difficulty in directly analyzing classification problems (under the cross-entropy loss).  Arguably the result of Theorem 5.2 may not perfectly explain the U-shaped curve in classification datasets, we however believe that they give adequate insight illuminating that scenario as well. Indeed, due to the difficulty analyzing the cross-entropy loss, most of the analytic works for deep learning study regression problems under the squared-error loss and use insights obtain this way to explain the behaviour of deep neural net in classification settings. See, for example,  Arora et al., Fine-Grained Analysis of Optimization and Generalization for Overparameterized Two-Layer Neural Networks, ICML 2019. Yang et al., Rethinking Bias-Variance Trade-off for Generalization of Neural Networks, ICML 2020.
>
> We have made an effort in the revision to better illustrate our logic flow and polish the writing for better clarity.
>
> >- On page 3: In Lemma 3.1, what is $\theta$ or what is a class of $\theta$? For example, can the authors write, "Then, for all $\theta\in$ ???"?
>
> The notation $\theta$ refers to a setting of the model parameter, which can be from any parameter space $\Theta$. Following your suggestion, we have added a line in the beginning of Section 3 and also polish the statement of Lemma 3.1 for clarity.
>
> >- On page 3: In Lemma 3.1, $\mathbb{E}_{\lambda}\hat{R}(\theta)$ is a random variable regarding the randomness of $\lambda$? That is, does the statement hold with probability one??
>
> In Lemma 3.1, fixing the training set $S$, the expression $\mathbb{E}_{\lambda}\hat{R}_\widetilde{S}(\theta)$ is not a random variable. For better clarity, we change the notation $\hat{R}_\widetilde{S}(\theta)$ to $\hat R _ { \widetilde S _ \lambda} (\theta)$ and rewrite the parts concerning this. Specifically, $\widetilde S _ \lambda$ is the synthetic dataset constructed by interpolating all pairs of examples in $S$ using interpolating parameter $\lambda$. Thus the randomness in $\hat R _  { \widetilde S _ \lambda}(\theta)$ is induced only by $\lambda$ when $S$ is fixed. Upon taking expectation,  $\mathbb E _\lambda \hat R _ { \widetilde S _ \lambda}(\theta)$ is deterministic. If the original training set $S$ were considered random, then $\mathbb E _\lambda \hat R _ { \widetilde S _ \lambda}(\theta)$ would be a random variable. In that case, conditioned on that $S$ is a balanced training set, the lower bound holds with probability 1. We hope the current version of the lemma has made this clear (where we specify that $S$ is fixed.)
>
> >- The figures are hard to read because the characters are very small. It is not kind to people who read the printed paper.
>
> We have enlarged the font sizes of the text in the figures.
>
> >- While the y-axis of the experimental results for test data in Figure 1 is test error, it is test accuracy in Figure 2. I read a printed paper, so I could not notice the change in the y-axis and was confused.
>
> Yes, in Figure 1 in Section 1 we measured the generalization performance by the test error, while in the figures in Section 3 we measured the generalization performance by the test accuracy. We have struggled in deciding whether to use the test error or the accuracy uniformly for all figures. However, we feel that in most of the context, it is more natural to display accuracy. On the other hand, using an accuracy curve, the "U"-shape would become "n"-shape, a term that is hardly used or understood. So we choose to only display test error in Introduction to show the U-shaped generalization behavior and leave all other figures displaying accuracy.
>
> >- On page 5: In Theorem 5.1, does the bound hold with probability one?
>
> We have revised the statement of theorem for better clarity, where we specify that $X$, $X'$  $\widetilde{X}$ and $\lambda$ are all fixed. In this setting, all quantities involved in the theorem are deterministic (i.e., not random variables).

---

> > ### Author Response · Authors · 2022-11-17
> > **To Reviewer VPXk (cont.)**
> >
> > >- On page 6: Is the definition that "For example, if $f$ is strongly convex with some parameter..." applied for all $x$?
> >
> > Yes. When saying that $f$ strongly convex, it is customarily implied that the condition holds for all $x$ in the domain of $f$.
> >
> > >- I omit other details but feel that the definition and statement in the theoretical analysis are not rigorous. I would like the authors to clarify them in more detail.
> >
> > We have made an effort to polish the presentation and clarify some confusions. If there is any other confusion, we would be glad to further clarify.
> >
> > >- On page 3: Can we interpreted $\mathcal{Y}$ as being $\mathcal{Y}\in\mathcal{P(Y)}$.
> >
> > When each label in ${\cal Y}$ is treated as a one-hot vector, it can be understood as a (degenerate) probability distribution over ${\cal Y}$, thus can be interpreted as elements of ${\cal P}({\cal Y})$.
> >
> > >- On page 3: In Lemma 3.1, what is the definition of $\mathbb{E}_\lambda$?
> >
> > For any function $g(\lambda)$ of $\lambda$, the notation
> > $\mathbb{E}_{\lambda} g(\lambda)$ is defined by:
> >
> > $\mathbb E _ \lambda g(\lambda)= \int _ \lambda g(\lambda)p(\lambda) d\lambda$
> > where $p(\lambda)$ is the distribution of $\lambda$. In the context of Lemma 3.1, $g(\lambda)$ is $\hat R _ {S_\lambda}(\theta)$ (under the revised notation) and $p(\lambda)$ is the Beta (1, 1) distribution.

---

### Author Response · Authors · 2022-11-17
**Response to All Reviewers**

Thank you very much for your comments and criticisms. We have revised the paper accordingly (revisions presented in red fonts, for the ease of the reviewer) and will address your individual comments separately. Below we address some concerns common to some reviewers.

### On Data Augmentation

In the revision (Appendix B.2), we added a discussion about data augmentation and presented additional experimental results.

In our view, when data augmentation is properly performed, i.e., not introducing examples with incorrect labels, it can be viewed as simply expanding the training set.  In this case, the impact of data augmentation on Mixup and its over-training dynamics is via the increase of training set. In Section 5.2 (Lemma 5.1 and Theorem 5.2),  it can be seen that as the size $m$ of the training set increases, the speed of exponential decay, say in both the first and second term of Equation (3), decreases. This will lead to the turning point of the U-shape curve to arrive later. This is also shown through the investigation of the impact of data size on the U-shaped curve in Section 7, where we have shown that larger size datasets may postpone the turning point of the U-shaped error curve. It is noteworthy that data augmentation drastically increases the training set, thus the over-training regime arrives significantly later.

To verify this, we have carried out a new series of experiments in which we overtrained ResNet18 on 10% CIFAR10 and CIFAR100 training sets with data augmentation; note that only using 10% of the training data is to reduce the number of training epochs required to observe the turning point. The results (measured with testing error) are provided in Appendix B.2, and they show a similar U-shaped curve, as had been observed in the previous experiments without data augmentation. This should validate our above discussion.

### On the Contribution of the Paper

We would like to note that the contributions of this work is primarily analytic rather than synthetic. Specifically,
1. We report a novel observation, where over-training with Mixup may compromise generalization by showing a U-shaped error curve.
2. We also explain the phenomenon theoretically, where we show that MixUp training introduces label noises and that  phenomenon is attributed to this noise. The more noise is introduced, the earlier will the turning point of the U curve occur.

Although it is of great interest to find better regularization schemes that fix this problem of MixUp, this has not been our focus. The presented solution (switching off Mixup) mainly serves as an additional support for our analysis, rather than building a SOTA MixUp-like scheme.

Having said these, we however argue that the observation and analysis of this work do provide benefit for practitioners. This is because MixUp has been widely adopted, and its numerous variants have been and are continuously being developed. The results of this paper alert the practitioners of the potential risk of Mixup over-training. In particular, when the mixing strategy results in aggressive revision of input examples and induce strong noise, the turning point of the U curve may arrive much earlier. Awareness of our results may provide guidelines for practical designs of new Mixup-like training algorithms. For example, one can not assume that Mixup won't overfit; cross-validation so as to find a good point to stop training is necessary.

---

### Decision · Program_Chairs · 2023-01-20

**Decision:**

Accept: poster

**Justification For Why Not Higher Score:**

Due to concerns related to impact of the paper.

**Justification For Why Not Lower Score:**

N/A

**Metareview: Summary, Strengths And Weaknesses:**

The paper exposes an interesting phenomenon occurring when training deep neural networks with mixup regularization referred to as over-training with mixup. After a certain number of epochs, the network might start to overfit to the noise introduced by mixup regularization.

Reviewers were divided in their opinions of the paper, with some leaning towards acceptance and others leaning towards rejection. However, there was good agreement on the pros and cons of the paper, and the main discussions revolved around the relative importance of each pro and con.

Overall, reviewers agreed that the phenomenon exposed in the paper is interesting and that the experiments sufficiently corroborate it. However, there was some concern about the practical impact of the paper, as the overtrain with mixup phenomenon seems to require an unrealistic number of epochs to occur, and therefore may not have a significant impact on performance in real-world scenarios. Additionally, there were concerns about the theoretical explanation provided in the paper, with some reviewers questioning how well it directly explains the observed experimental results.

Despite these concerns, reviewers were hopeful that the paper would stimulate further work in this direction, both practical and theoretical. Based on this grounds, it is my pleasure to recommend acceptance of the paper.

Please remember to address the reviewers' remarks. On top of that, please bring into the main text discussion and results related to overtraining in training with augmentation, and further elaborate on how the theory explains the observed experimental results.

**Note From Pc:**

if the above contains the word "oral" or "spotlight" please see: "oral" presentation means -> notable-top-5% and "spotlight" means -> notable-top-25%. As stated in our emails, we are disassociating presentation type from AC recommendations

**Summary Of Ac-Reviewer Meeting:**

We discussed pros and cons of the paper. We agreed on the set of pros and cons and chatted a bit about potential impact of the paper. We then discussed that reviewers on the call would be fine with both accepting and rejecting the paper.